# BAOS-CNN: A novel deep neuroevolution algorithm for multispecies seagrass detection

Md Kislu Noman ●*❦, Syed Mohammed Shamsul Islam ●❦, Seyed Mohammad Jafar Jalali❦, Jumana Abu-Khalaf, Paul Lavery

School of Science, Edith Cowan University, Perth, Australia

❦ These authors contributed equally to this work.
* md.k.noman@gmail.com

**Data Availability Statement:** This paper's multi-species seagrass dataset is available at: https://ro.ecu.edu.au/datasets/141/.

## Abstract

Deep learning, a subset of machine learning that utilizes neural networks, has seen significant advancements in recent years. These advancements have led to breakthroughs in a wide range of fields, from natural language processing to computer vision, and have the potential to revolutionize many industries or organizations. They have also demonstrated exceptional performance in the identification and mapping of seagrass images. However, these deep learning models, particularly the popular Convolutional Neural Networks (CNNs) require architectural engineering and hyperparameter tuning. This paper proposes a Deep Neuroevolutionary (DNE) model that can automate the architectural engineering and hyperparameter tuning of CNNs models by developing and using a novel metaheuristic algorithm, named 'Boosted Atomic Orbital Search (BAOS)'. The proposed BAOS is an improved version of the recently proposed Atomic Orbital Search (AOS) algorithm which is based on the principle of atomic model and quantum mechanics. The proposed algorithm leverages the power of the Lévy flight technique to boost the performance of the AOS algorithm. The proposed DNE algorithm (BAOS-CNN) is trained, evaluated and compared with six popular optimisation algorithms on a patch-based multi-species seagrass dataset. This proposed BAOS-CNN model achieves the highest overall accuracy (97.48%) among the seven evolutionary-based CNN models. The proposed model also achieves the state-of-the-art overall accuracy of 92.30% and 93.5% on the publicly available four classes and five classes version of the 'DeepSeagrass' dataset, respectively. This multi-species seagrass dataset is available at: https://ro.ecu.edu.au/datasets/141/.

## 1. Introduction

In recent years, deep learning-based techniques have achieved enormous success in the field of artificial intelligence (AI). Now research communities are extensively using deep learning models in a variety of scientific and industrial applications [1, 2]. The Convolutional Neural Networks (CNNs)-based models are gaining popularity due to their higher accuracy, robustness and automatic feature extraction ability [3, 4] and providing almost human-level precision in many applications [5, 6]. The CNNs-based deep learning models can extract the

**Funding:** This research work is supported by the Australian Government Research Training Program (RTP) scholarship, and the funds were received by Dr. Kislu Noman. The funders had no role in study design, data collection and analysis, decision to publish, or preparation of the manuscript.

**Competing interests:** The authors have declared that no competing interests exist.

important key features automatically from the input data, lessening the need for manual, hardcore feature extraction as needed in traditional machine learning approaches [7]. The CNNs-based deep learning approaches have shown better performance over machine learning techniques for identifying seagrass or seagrass meadows in underwater imagery [8–10]. For instance, Reus et al. [8] showed that the deep learning-based CNNs model achieved the highest accuracy of 94.5% over Histogram of Oriented Gradients (HOG) and Local Binary Patterns (LBP)-based machine learning techniques. Perez et al. [9] showed that deep learning-based models (specifically, the deep capsule network (DCN) and the deep convolutional neural network (CNN)) had significantly better performance identifying seagrass than machine learning-based models (namely, linear regression and support vector machine-based (SVM)) on the WorldView-2 satellite imagery dataset. Gonzalez-Cid et al. [10] also reported that a CNNs-based deep learning model achieved 1.54% better accuracy identifying *Posidonia Oceanica* than the SVM-based machine learning model on their colour image datasets. They conducted their experiments on two datasets: dataset 1 contained 69 and dataset 2 contained an additional 180 colour images. They captured these images by using an Autonomous Underwater Vehicle (AUV) with a bottom-looking camera in several coastal areas of Mallorca, Spain.

Due to the improved performance noted above, most of the recent seagrass identification or mapping approaches have applied deep learning-based models. Martin-Abadal et al. [11] developed a semantic segmentation model (VGG16-FCN8 model) to identify *Posidonia Oceanica* meadow and achieved a precision of 96.57% and accuracy of 96.81%. Weidmann et al. [12] applied four deep learning techniques (DeepLabv3+, Fully Convolutional Network (FCN), Dilated Convolution Network (dilNet) and U-Net) on the 'LookingForSeagrass' dataset. They captured 12682 images using an AUV along the coast of the island of Murter, Croatia. The annotated dataset consists of 6037 annotated images. These images were annotated with polygons that define a pixel belonging to the seagrass or background. They achieved the highest 87.78% mIoU (mean Intersection over Union) by applying the DeepLabV3+ model. Wang et al. 2020 [13] mapped seagrass by implementing an encoder-decoder-based segmentation model on the same dataset and achieved a mIoU of 90.66%. Noman et al. [14] proposed a Faster R-CNN network with NASNet backbone and achieved an mAP of 0.412 on the *Halophila ovalis* dataset. Recently, Noman et al. [15] achieved the highest mAP of 0.484 on the same dataset by implementing the EfficientDet-D7–based seagrass detector. The aforementioned two approaches were developed to detect only a single *Halophila ovalis* seagrass species.

In 2020, Raine et al. [16] first identified multi-species seagrass from underwater digital images by applying transfer learning-based VGG-16 and ResNet-50 models on their patch-based 'DeepSeagrass' dataset (This dataset is described in Section 3.2). Their VGG-16-based classifier achieved an overall accuracy of 88.2% on the four classes version of the dataset. They achieved an overall accuracy of 92.4% on their five classes version of the dataset. Recently, Noman et al. [17] proposed a transfer learning-based EfficientNet-B5 classifier and achieved 91.3% and 93.0% overall accuracies on four classes and five classes versions of the 'DeepSeagrass' dataset respectively. Noman et al. [17] also developed a two-stage semi-supervised deep learning model with an EfficientNet-B5 backbone network. This technique was able to label a large dataset automatically with the guidance of a small labelled multi-species seagrass dataset. This patch-based two-stage architecture achieved an accuracy of 98.0% on their own 'ECU-MSS' dataset.

From the above review of the literature, we can see that there are only two deep learning-based approaches so far proposed for multi-species seagrass classification tasks from underwater digital images. Both of these approaches are based on the transfer learning technique. The main limitation of the transfer learning technique is negative transfer. Negative transfer occurs when the initial and target problems are not similar enough [18]. Backpropagation techniques

are frequently applied to develop a CNN model. The backpropagation techniques can initialize the weights of the convolution and fully connected layers [19, 20]. However, backpropagation-based models get stuck in local minima during the training time. On the other hand, CNNs training from scratch requires architectural engineering and hyperparameter tuning. Hyper-parameter optimisation of CNNs is one of the most challenging tasks. Small variations in the hyperparameter values can influence the overall performance of the model [21, 22]. The Deep Neuroevolutionary algorithms (DNEs) are practical solutions that optimise both the architecture and hyperparameters of the CNNs model to find out the best architecture design and hyperparameters set [23, 24]. This algorithm can reduce the overfitting and the local minima problem in developing CNNs models from scratch [25]. They can automate the neural network architecture by applying evolutionary algorithms or metaheuristic algorithms [23, 26].

A metaheuristic algorithm is a general-purpose stochastic heuristic algorithm, which is frequently used to solve almost all kinds of optimisation problems [27, 28]. The Atomic Orbital Search (AOS) algorithm is a recent metaheuristic approach that has demonstrated exceptional performance in addressing engineering design and mathematical issues. The AOS algorithm is capable of dealing with computational complexity issues by getting better results with lower functional evaluations. To strengthen the global search for the original version of AOS, we combined Lévy flight with the AOS. The Lévy flight is a generalized Brownian motion whose individual jumps' lengths are distributed with the non-Gaussian random distribution function [29]. Lévy flight technique improved the searchability of many metaheuristic algorithms [30, 31]. For instance, Hakli and Ŭguz [32] improved the Particle swarm optimisation algorithm, Yang and Deb [33] created a new Cuckoo in Cuckoo search algorithm and Yang [34] introduced a new version of Firefly Algorithm with Levi flight technique. The Lévy flight technique was also used to diversify the ant colony optimisation algorithm [35, 36].

Deep Neuroevolutionary algorithms combine the power of deep neural networks (which excel at learning complex patterns) and representations, with neuroevolutionary algorithms that optimize neural network architectures and parameters using evolutionary principles. Ijjina et al. [37] proposed a neural network classifier that utilized the power of a genetic algorithm to optimize the initial weights. Their approach mitigated the challenges of overfitting and local minimum issues. They achieved an accuracy of 99.98% in human action recognition on the UCF50 dataset. Martin et al. [38] proposed the EvoDeep algorithm, a deep neuroevolutionary algorithm to optimize both the parameters and architecture of Deep Neural Networks (DNNs). Their approach resulted in an accuracy of 98.93% on the handwritten digit dataset. Sun et al. [39] proposed the EvoCNN algorithm using genetic algorithms for evolving the architectures and connection weight initialization values of a deep convolutional neural network to address image classification problems. Davoudi et al. (2021) proposed a deep neuroevolutionary algorithm powered by genetic algorithms (GA) for the classification of Breast Cancer images. Their method outperformed the traditional Adam optimizer, achieving an image classification accuracy of 85%. Sajad et al. [40] proposed an enhanced variant of the Boosted Salp Swarm Algorithm (BSSA) to improve the original version of the Salp Swarm Algorithm (SSA). This enhancement involved the integration of two powerful optimization operators: opposition-based learning (OBL) and chaotic maps. Their two-stage deep neuroevolution (DNE) framework was developed to enhance the classification of chest X-ray images. By applying deep neuroevolutionary techniques, researchers aim to enhance the performance, efficiency, and adaptability of deep learning models. Jalali et al. [41] proposed an evolutionary algorithm by improving the basic version of competitive swarm optimizer by using three powerful evolutionary operators: Cauchy Mutation (CM), Evolutionary Boundary Constraint Handling (EBCH), and tent chaotic map. Their deep neuroevolutionary algorithm automated the identification process of COVID-19 disease from chest X-ray images, achieving an average

accuracy of 98.56%. By reviewing the literature, we can see a significant advancement and immense potential in developing and hyper-tuning image classification models.

In this paper, we develop a novel metaheuristic algorithm by incorporating the Lévy flight technique [42] in the AOS algorithm [43]. We name the novel optimisation algorithm as Boosted Atomic Orbital Search (BAOS). We then apply the proposed algorithm on a multi-species seagrass dataset to automate the design of the architecture and choose the best hyper-parameter set of that CNNs model. We propose this algorithm to find out a solution in a search space quickly and accurately. We also want to improve the search technique of the AOS algorithm by balancing the exploration and exploitation in the search space. Thus, the proposed algorithm can minimize the requirement of manual hyperparameter tuning and can develop CNNs architecture without being an expert in the field of AI.

The main contributions of this paper can be summarised as:

- A novel optimisation algorithm is proposed by incorporating the Lévy flight technique with the Atomic Orbital Search algorithm.

- The proposed BAOS algorithm can automate the hyperparameter tuning and architectural engineering and minimize the requirement of manual tuning during the evolutionary search.

- This proposed DNE framework (BAOS-CNN) outperforms six other popular optimisations algorithm-based DNE frameworks.

- The proposed DNE framework (BAOS-CNN) outperforms the state-of-the-art multi-species seagrass classifiers.

- A large patch-based dataset is developed for the multi-species seagrass mapping task.

The remainder of the paper is organized as follows: Section 2 describes the Methodology. Datasets are presented in Section 3, while Section 3.3 presents the performance evaluation of our proposed model. Finally, Section 4 presents the conclusions.

## 2. Methodology

In this section, we first describe the basics of CNNs architecture, AOS algorithm and Lévy flight technique. Then we describe the development of a novel evolutionary algorithm. Next, we describe the development of a novel DNE to tune the hyperparameters and architectural design of CNNs. Finally, we describe related evaluation metrics.

### 2.1 CNNs architecture

Convolutional Neural Networks (CNNs) are the most common and popular deep learning architectures based on visual perception mechanisms [44]. CNNs are deep learning architectures that extract information directly from the data. There is a variety of CNNs architectures available. However, the basic components of CNNs are very similar. Convolutional, pooling, and fully connected layers make up the CNNs model [45, 46]. A convolutional layer learns feature representations of the input data automatically. A feature map connects each neuron with its neighbours in the previous layer. CNNs use kernels or filters over the images to produce the feature maps. To produce the feature map, the input image can first be convolved with a kernel, followed by applying a nonlinear activation function on the convolved results [47, 48]. Feature maps are generated using common kernels for all spatial locations in the input images. Several kernels are used to obtain the complete feature maps [49]. The lower-level layers are used to detect simpler patterns or to extract low-level features (edges, curves etc.) and higher-

level layers are used to encode more abstract features or complex patterns (faces, objects, etc). In order to extract higher-level features or information from the input data, several layers of convolutional and pooling operations are typically stacked together. These layers work together to process the input data, with each subsequent layer building upon the features learned by the previous layer. The convolutional layers are responsible for detecting patterns and features in the input data, while the pooling layers are responsible for reducing the dimensionality of the data and increasing the robustness of the features learned. Together, these layers form the backbone of a convolutional neural network (CNN) and are crucial for achieving good performance on tasks such as image recognition, object detection, and natural language processing.

The feature map at location *(i,j)* in the $k^{th}$ feature map of $l^{th}$ layer, $z^l_{i,j,k}$, cab be mathematically defined as per [50]:

$$z^l_{i,j,k} = w^{iT}_k x^l_{j,j} + b^l_k \qquad (1)$$

where $w^{iT}_k$ is the weight vector and $b^l_k$ is the bias of the $k^{th}$ filter of the $l^{th}$ layer. $X^l_{i,j}$ represents the input patch at location *(i, j)* for the $l^{th}$ layer.

Activation functions decide whether a neuron is to be activated or not. In other words, it will determine whether the neuron's input to the network matters during prediction [51]. The activation function provides nonlinearities to CNNs models that are ideal for detecting nonlinear features. The activation $a^l_{i,j,k}$ of the convolutional feature $z^l_{i,j,k}$ can be calculated as:

$$a^l_{i,j,k} = a\left(z^l_{i,j,k}\right) \qquad (2)$$

Convolutional feature maps are summed up by pooling layers. The dimension of the feature maps is reduced by pooling layers [52]. Consequently, the network has fewer parameters to learn and performs fewer computations. Usually, a pooling layer is placed between two convolutional layers. In the pooling layer, the corresponding feature maps of the preceding convolutional layer are connected together. The pooling layer can reduce the dimension of the feature map $a^l_{m,n,k}$ by

$$y^l_{i,j,k} = pool(a^l_{m,n,j}, \forall (m, n) \in R_{i,j} \qquad (3)$$

where the location *(i, j)* and local neighbourhood location is $R_{i,j}$.

The fully connected layer is the last in the CNNs. High-level feature reasoning is performed using fully connected layers. In this layer, neurons are fully connected to neurons in previous and succeeding layers [53]. Finally, the Softmax activation function is applied at the end of CNNs to produce the final output [54].

## 2.2 Evolutionary algorithm: Atomic Orbital Search (AOS)

The Atomic Orbital Search (AOS) optimization algorithm is built on the principle of the atomic orbital model [43]. The algorithm is based on the behaviour of electrons in an atom, which occupies certain energy levels and moves in specific orbits. In AOS, the optimization problem is represented as an atomic system, where the electrons represent the decision variables and the energy levels correspond to the cost function. The electrons move in orbits around the nucleus, which represents the current best solution. The algorithm uses a quantum-inspired mechanism to control the electrons' movement and transition between energy levels, in order to explore the search space and find the optimal solution.

The AOS algorithm is inspired by quantum mechanics and it's a new optimization algorithm that presents a good performance on various optimization problems. The algorithm is

inspired by the behaviour of electrons in an atom, which occupies certain energy levels and moves in specific orbits. AOS uses a quantum-inspired mechanism to control the electrons' movement and transition between energy levels, in order to explore the search space and find the optimal solution. Based on a quantum-based atomic model, AOS considers several candidates *(X)* for representing electrons around the nucleus. As a cloud electron surrounding the nucleus, the search space of this algorithm is described as thin, concentric, spherical layers. In the search space, each electron represents a candidate solution *(Xi)*. The mathematical equation for the *m* candidate solutions in a *d*-dimension space is:

$$X = \begin{bmatrix} X_1 \\ X_2 \\ . \\ . \\ X_i \\ . \\ . \\ X_m \end{bmatrix} = \begin{bmatrix} X_1^1 & X_1^2 & . & . & . & X_1^j & . & . & . & X_1^d \\ X_2^1 & X_2^2 & . & . & . & X_2^j & . & . & . & X_2^d \\ . & . & . & . & . & . & . & . & . & . \\ . & . & . & . & . & . & . & . & . & . \\ X_i^1 & X_i^2 & . & . & . & X_i^j & . & . & . & X_i^d \\ . & . & . & . & . & . & . & . & . & . \\ . & . & . & . & . & . & . & . & . & . \\ X_m^1 & X_n^2 & . & . & . & X_m^j & . & . & . & X_m^d \end{bmatrix} \tag{4}$$

where *i = 1, 2,. . ., m* and *j = 1, 2,. . ., d*. The number of solutions within the electron cloud is *m*. The problem dimension *d* represents the position of the electrons or candidates. The positions of electrons within the electron cloud are randomly determined using the following mathematical equation:

$$X_i^j(0) = X_{j,min}^i + rand.\left(X_{i,max}^j - X_{i,min}^j\right) \tag{5}$$

where *i = 1, 2,. . ., m* and *j = 1, 2,. . ., d*. The initial position of the candidates is represented by $X_i^j(0)$. The minimum and maximum bounds of the $j^{th}$ decision variable for the $i^{th}$ solution candidate are represented by $X_{i,min}^j$ and $X_{i,max}^j$, respectively. The variable *Rand* is a uniformly distributed integer that ranges from 0 to 1.

An electron has an energy state that determines its objective function value in the mathematical model. In this mathematical model, solution candidates represent electrons at lower energy levels for higher objective function values. In contrast, the solution candidates represent electrons at higher energy levels for lower objective function values. In order to contain the objective function values (energy levels) of all the different solution candidates, the vector equation is used as follows:

$$X = \begin{bmatrix} E_1 \\ E_2 \\ . \\ . \\ E_i \\ . \\ . \\ E_m \end{bmatrix}, \ i = 1, 2, \dots, m \tag{6}$$

In this equation, $E_i$ denotes the energy level of the $i^{th}$ solution candidate, and $m$ represents the total number of electrons in the search space.

## 2.3 Lévy flight

Paul Pierre Lévy was a French mathematician who made significant contributions to the field of probability theory, and his discovery of Lévy stable distribution is an important addition to the field of probability theory and its applications [42]. Lévy found that the stationary increments of a class of non-Gaussian random processes, known as Lévy processes, were distributed according to a Lévy stable distribution. This means that, for a Lévy process, the distribution of the increments does not depend on the starting point of the process, but only on the time interval between the increments. The Lévy flight is also known as the Lévy motion. The Lévy flight is a type of random walk that is characterized by long jumps interspersed with short steps. The Lévy flight is a specific case of a Lévy process, which is a class of non-Gaussian random processes. In a Lévy flight, the lengths of the jumps are distributed according to a power-law distribution, also known as a Lévy-stable distribution. This means that the probability of a jump of length x is inversely proportional to x to the power of a parameter known as the Lévy index. The Lévy index determines the tail of the distribution, and it is used to classify Lévy flights into different types. Lévy stable distribution can improve the accuracy of an optimisation algorithm [32]. Lévy stable laws are important for three fundamental properties: Like the Gaussian law, the sums of random variables are attracted to Lévy stable laws [55]. Lévy flights are random walks whose step lengths have a heavy-tailed probability distribution, also called a Lévy distribution. The 2D Lévy flight distribution in 500 steps is presented in Fig 1. The Lévy distribution can be presented by a simple power-low formula $L(s) \sim |s|-1-\beta$ where β is in between 0 to 2 [33]. Then Lévy flight $L(s, \gamma, \mu)$ can be defined as:

$$L(s, \gamma, \mu) = \left\{ \sqrt{\frac{\gamma}{2}} \exp\left[ -\frac{\gamma}{2(s-\mu)} \right] \frac{1}{(s-\mu)^{\frac{3}{2}}} \; if \; 0 < \mu < s < \infty \; and \; s \leq 0 \quad (7) \right.$$

Here μ > 0 is the minimum step and γ is a scale parameter. Clearly, as s −∞, we have

$$L(s, \gamma, \mu) \approx \sqrt{\frac{\gamma}{2\pi}} \frac{1}{s^{\frac{3}{2}}} \quad (8)$$

This is a special case of the generalized Lévy distribution.

## 2.4 Proposed Lévy flight-based Atomic Orbital Search algorithm

We incorporate the Lévy flight technique into the original version of the Atomic Orbital Search (AOS) algorithm to make a more efficient and quicker AOS. We name the proposed Lévy flight-based AOS algorithm as Boosted Atomic Orbital Search (BAOS). The flowchart of the proposed BAOS algorithm is presented in Fig 2.

Quantum-based atomic models state that electrons with different energy levels can move between different layers with different energy levels around the nucleus. This mathematical model considers two positions updating processes for solution candidates. The primary consideration in the process is the interaction of photons with electrons, while other factors such as interactions with particles or magnetic fields are considered as secondary factors. Depending on the characteristics of the candidate and layer, these factors are evaluated for each solution candidate in the imagined layers. Photon Rate (PR) is used as a measure of the likelihood of photon-electron interactions. In AOS, the fitness value of PR is updated as follows: if the

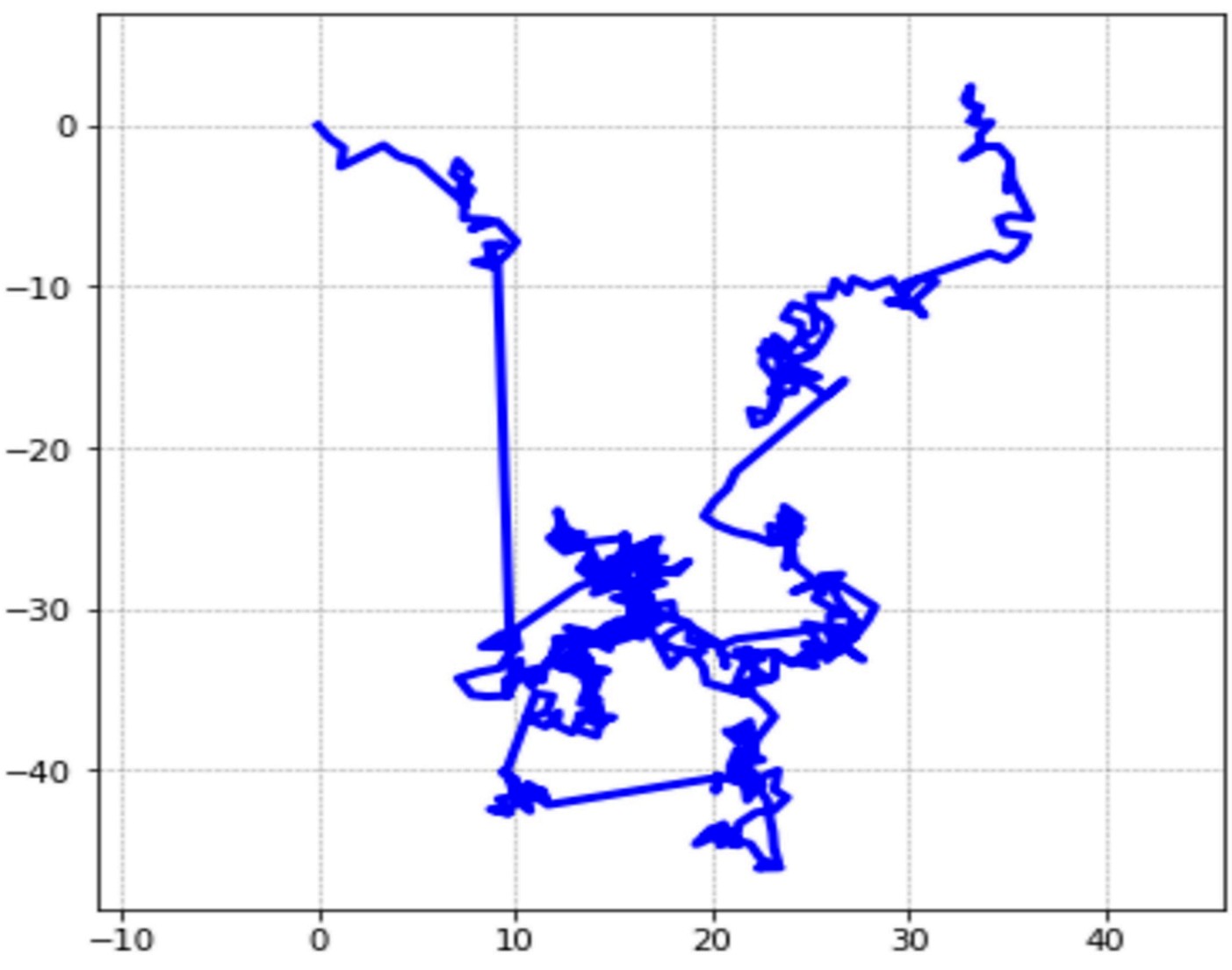

**Fig 1. 2D Lévy flights in 500 steps.**

calculated fitness value is better than the previous fitness value of PR, the calculated fitness value will be assigned as the new fitness value, and the fitness value's limit value will be reset. Otherwise, the fitness value's limit value will be incremented by 1. Similarly, if the updated fitness value of PR is smaller (to minimize the problem), the fitness value is set as the current value of PR. This process is repeated until the number of iterations reaches the maximum iteration count. The Lévy flight technique is incorporated into the AOS by

- Updating the Photon Rate (PR) for the solution in candidate in search space by using Lévy distribution method.

- For every PR, a limit value is set, which is incremented every iteration if the particles cannot improve themselves.

This new version of AOS can be able to improve the global search and can minimize the local minima problem. The new state of the PR is calculated using Lévy flight-based technique

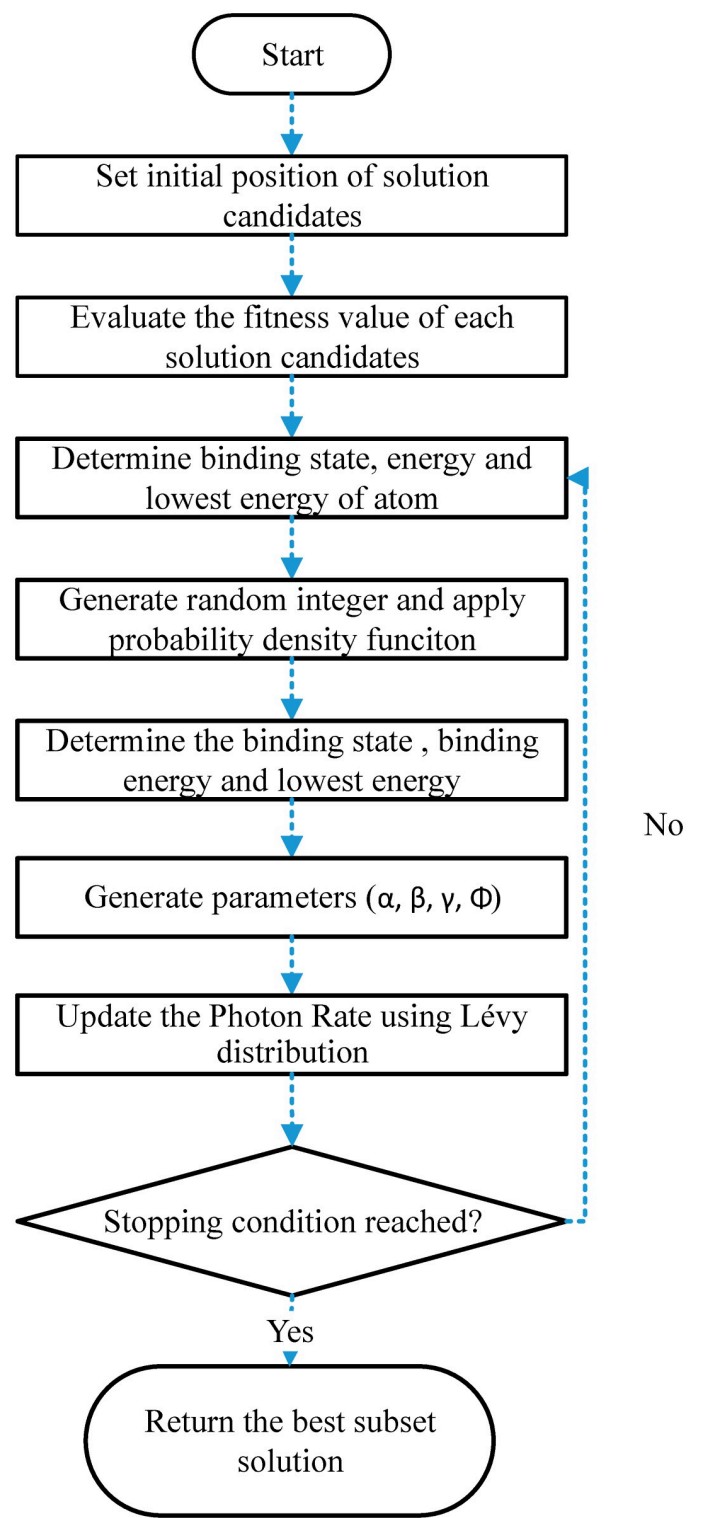

**Fig 2. Flowchart of the proposed BAOS algorithm.**

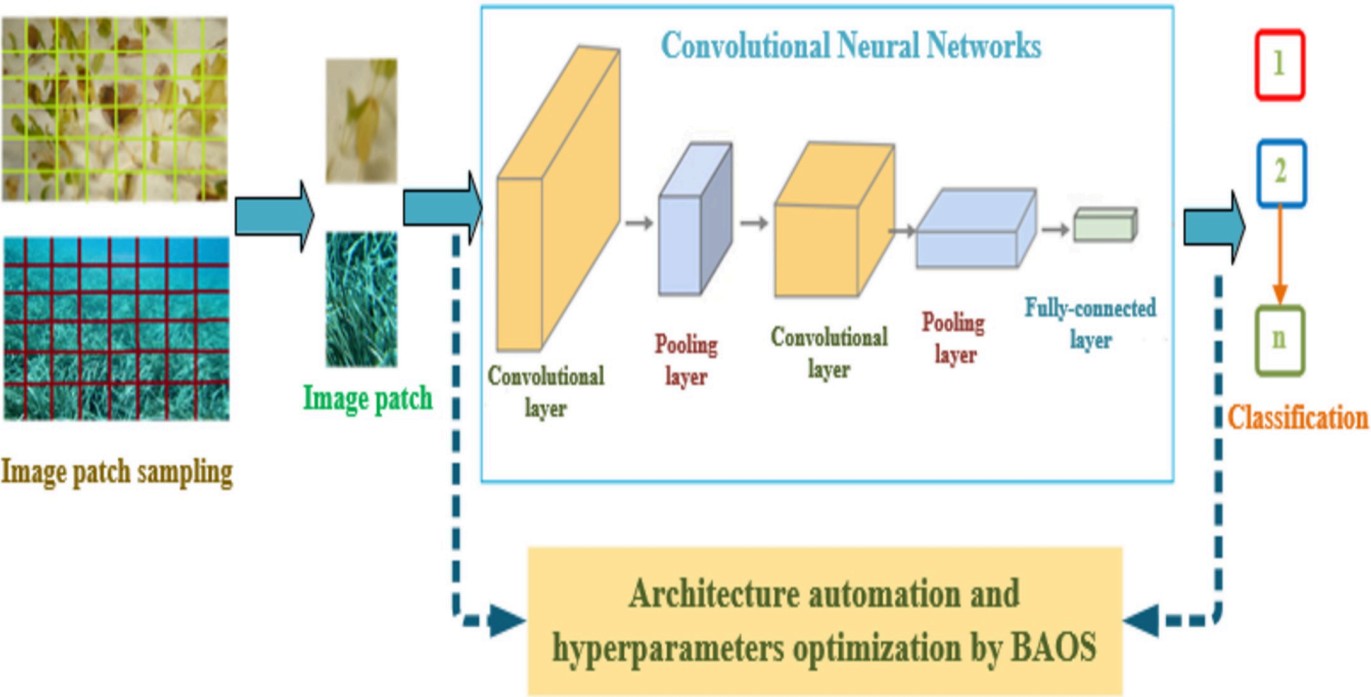

**Fig 3. The overall architecture of the proposed BAOS-CNN model.**

as

$$X^{t+1} = X^t + \alpha \oplus L\acute{e}vy(\beta)\,\alpha \tag{9}$$

where $\alpha$ is the step size and $\beta$ is the distribution corresponding to Gaussian distribution.

$$X^{t+1} = X^t + random(size(D)) \oplus L\acute{e}vy(\beta)\,\alpha \tag{10}$$

The product $\oplus$ means entry-wise multiplications. The Lévy flight technique enhances population diversity in the search space. This population diversity provides better exploration technique in the area of search dimension. This technique also enhances the exploitation in the search space. This exploration makes the proposed algorithm more efficient and quicker in the search space.

## 2.5 Proposed deep neuroevolutionary algorithm (BAOS-CNN)

The basic idea of DNE is to use an evolutionary algorithm to search for the best set of weights and biases for a neural network. This is done by creating a population of candidate solutions (CNNs) and then repeatedly applying selection, crossover, and mutation operators to generate new generations of solutions. The fitness of each network is evaluated using a performance metric such as accuracy on a validation set. The process is repeated until the performance of the best network reaches a satisfactory level or a stopping criterion is met. We propose utilizing a Deep NeuroEvolution (DNE) algorithm for the task of detecting multiple species of seagrass. The overall architecture of the proposed BAOS-CNN framework is presented in Fig 3. The

objective function 302 of DNE is defined as follows:

$$Accuracy = CNNs(\overrightarrow{H_p}, \overrightarrow{W_p}, Td_i) \tag{11}$$

where Eq 11 presents a function making the CNNs models from scratch. $\overrightarrow{H_p}$ represents hyperparameters, $\overrightarrow{W_p}$ represent weight vector and $Td_i$ indicates training dataset.

In the BAOS algorithm, the optimization problem is represented as an atomic system, with the electrons representing the decision variables, and the energy levels corresponding to the cost function.

To optimize and compute the fitness value for updating the hyperparameters of a CNN using BAOS algorithm, the following eight basic steps are followed:

1. **Initialize Hyperparameters**: The first step is to initialize the electrons, which represent the different sets of parameters for the CNN. Each electron is randomly assigned an energy level and an orbital position. These hyperparameters will be part of the solution space that the evolutionary algorithm will explore.

2. **Encoding**: Encode the hyperparameters into a suitable representation that the BAOS Algorithm can work with. The proposed BAOS helps to determine the binding energy (BE) and the binding state (BS) of the atom and finds out the candidate with the lowest energy level (LE). Each solution represents one of nine hyperparameters with a nine-dimensional vector $X_{ij}$, where $i = 1, 2, \ldots, n$ and each hyperparameter is $j = 1, 2, \ldots, 9$.

3. **Solution Generation**: Generate initial solutions by randomly selecting or generating hyperparameter configurations within the defined solution space. Each solution corresponds to a specific set of hyperparameters for the CNN.

4. **Solution Update**: The fourth step is to update the electrons' energy levels and positions with Lévy distribution based on their current positions and the positions of their neighbouring electrons. The electrons' positions are updated using a quantum-inspired mechanism that allows them to jump to different energy levels and explore the search space.

5. **Fitness Evaluation**: This step involves the performance evaluation of the CNN based on the electron's position in the orbital space. This can be done by training the CNN with the corresponding set of parameters and measuring its performance using performance metrics such as accuracy or mean squared error. During training, the hyperparameters are used to construct and configure the CNN model. The training process involves feeding the dataset through the CNN, and updating the network's hyperparameters and weights. The final step involves assessing the performance of the CNN on the specified evaluation metrics.

6. **Fitness Calculation**: Assign a fitness value to each solution based on the CNN's performance on the evaluation metrics. Higher fitness values are assigned to solutions that result in better performance. The fitness value can be determined directly from the evaluation metrics. The evolutionary algorithm can explore and identify hyperparameter settings during the training time on the dataset.

7. **Selection**: The best set of parameters is selected based on the corresponding electrons with the lowest energy levels.

8. **Repeat**: By iteratively applying these steps within the BAOS algorithm, the hyperparameters of the CNN are optimized and updated based on their fitness values. This process allows for the identification of hyperparameter configurations that improve the CNN's performance

on the chosen evaluation metrics. The process of evaluation, update, and selection is repeated until the optimal solution is found.

CNNs architecture (Section 2.1) explains that CNNs are constructed with convolution layers, pooling layers, dropout layers, and fully connected layers. The proposed BAOS algorithm is applied to optimise the hyperparameters, depth and width of the CNNs layers.

A Probability Density Function (PDF) is a mathematical function that describes the probability distribution of a continuous random variable. It gives the probability of a value of the random variable to occur. It's a fundamental concept in probability theory and statistics and it's widely used to model various phenomena in different fields. In AOS, the PDF determines the position of electrons around the nucleus. The PDF represents a variable's likelihood of electron position within a given range, based on probability theory. The representation of solutions and the calculation of fitness function are considered in our proposed model. Learning rate, dropout rate, and momentum rate are the hyperparameters with continuous values. The proposed BAOS algorithm can identify the optimal values of those hyperparameters directly. The number of convolutional layers, number of filters, filter size, number of epochs, batch size, and max-pooling size are other hyperparameters with discrete values. Discrete values of these hyperparameters need to be transformed into their respective optimal values. The values are transformed into integers using an efficient model [40]. Finally, the model transforms the continuous value of each hyperparameter into $D = [K1,K2,...,Kn]$ as a discrete search space. The discretization model is:

$$\alpha = 1 + n \times R \tag{12}$$

$$\beta = min(\lfloor \alpha \rfloor, n) \tag{13}$$

where R is a real value to explore in the continuous search space in a range of *[0,1]*, $\alpha$ maps *R* to *[1,n + 1]* and $\beta$ maps from $\alpha$ to *[1,2,...,n]*. Next, each solution can be calculated by taking the integer value corresponding to its continuous dimension.

$$Xij = K\beta \tag{14}$$

The proposed BAOS algorithm randomly initializes the position of solution candidates *(Xij)* in the area of the search space with m candidates. This algorithm finds fitness value (Ei) for initial solution candidates. The proposed algorithm finds out the optimal solution by continuously upgrading current solutions. The steps of the proposed BAOS algorithm are presented in Algorithm 1.

**Algorithm 1** Proposed DNE (BAOS-CNN) based Atomic Orbital Search algorithm with Lévy flight technique
1. Set Initial population of solution candidates *(Xi)*
2. Evaluate the fitness values *(Ei)*, determine the binding state *(BS)*, binding energy *(BE)* and lowest energy *(LE)* and generate a random number (n)
3. Apply Probability Density Function *(PDF)* to find out the binding state and binding energy
4. Update the random parameters *(α, β, γ, φ)*
5. Determine the Photon Rate PR and update based on Lévy distribution
6. Set initial population *Xi (i = 1,2,,... .,P)*;
7. Set *t = 1;*
8. while (*t < Max_{ite}*) do
9. Set Bests = the best obtained Lévy distribution;
10.   For each *PR Xi* do
11.   Update the parameter *C2*

```
12.   if Xi leader for PR then update the leader PR position
13.   else if Xi followers then Update the PR
14.   end if
15.   Repair and check the PR if exceeded the search boundaries
16.   Update the Best_S;
17.   end for
18.   Set t = t+1
19.   end while
20.   Adjust the CNNs model based on the hyperparameters obtained from
      Lévy distribution-based AOS
21.   Classify the images in the test set using the best CNNs model
```

The proposed BAOS-CNN algorithm combines the Atomic Orbital Search (AOS) algorithm with the Lévy flight technique to optimize the hyperparameters of convolutional neural networks (CNNs). The algorithm starts by initializing a population of solution candidates representing different hyperparameter configurations for the CNNs. Fitness values are evaluated, and binding states and binding energies are determined based on these fitness values. The algorithm then applies a Probability Density Function to identify promising solutions. Random parameters are updated, and the Photon Rate (PR) is calculated and adjusted using the Lévy distribution. The algorithm iterates through the population, updating parameters, and positions, and checking boundaries. This algorithm follows an iterative process to search for optimal hyperparameters. The algorithm initializes a population of solution candidates and evaluates their fitness values, identifying binding states and energies. Through the iteration loop, the algorithm updates parameters and positions based on specific rules and calculates the Photon Rate (PR) using the Lévy distribution. Repair and boundary checks are conducted to ensure valid solutions. The best solution, as determined by the Lévy distribution, is then used to adjust the CNNs model. Finally, the adjusted CNNs model is applied to classify images in the test set, completing the algorithm's optimization process for hyperparameter selection in CNNs.

The BAOS-CNN model is based on quantum mechanics and states that electrons with varying energy levels can transition between layers with differing energy levels. This model presents a mathematical representation that combines two processes for updating solution candidates, where the main process is the interaction with electrons by photons, while the secondary process is the interaction with particles or magnetic fields by particles or magnetic fields. These processes are conducted for each candidate within the imaginary created layers according to their characteristics. Each electron is assigned a uniformly distributed random number $\emptyset$ in the range of (0, 1) to represent the effect of photons on it. PR is an update based on the Lévy flight random distribution in the search space. The list of all hyperparameters that are optimised for our proposed model is presented in Table 1.

We also optimised the same CNNs model with six other popular optimisation algorithms: Grey Wolf Optimiser (GWO) [56], Salp Swarm Algorithm (SSA) [57], Particle Swarm Optimisation (PSO) [58], Foth Flame Optimisation (MFO) [59], Chaotic Salp Swarm Algorithm (CSSA) [60] and Atomic orbital search (AOS) [43]. All the DNEs algorithms (GWO-CNN, SSA-CNN, PSO-CNN, MFO-CNN, CSSA-CNN, AOS-CNN and BAOS-CNN) were trained and tested on a multi-species 'ECU-MSS-2' dataset. First, the dataset was divided into training and test set. All the DNEs were trained using the K-fold cross-validation technique. The value of K was set to five for all the DNEs. In five-fold cross-validation, the training dataset was partitioned into five sub-folds. The four sub-folds were selected as the training set and the remaining sub-fold was selected as a validation set. The process was repeated five times until each group was treated as a validation set and the remaining as the training set. To ensure that the class distribution was consistent between the training and validation sets, we used the Stratified

**Table 1. List of CNNs hyperparameters for our proposed model.**

| Hyperparameters | Symbol | Value |
|---|---|---|
| Number of Convolutional Layers | Nc | [1, 2,. . .., 20] |
| Filter size | Ks | [1, . . .. . ., 30] |
| Number of filters | Nf | [1, . . .. . ., 500] |
| Number of epochs | Ne | [1, . . .. . ., 400] |
| Batch size | Bs | [10, 20,. . .., 200] |
| Max-pooling size | MPs | [1, . . .. . .... . ., 20] |
| Dropout | Dr | [0.2, 0.25,. . .., 0.65] |
| Learning rate | Lr | [0.001, 0.006,. . .., 0.1] |
| Momentum rate | Mr | [0.05, 0.1, . . .. . ...., 0.95] |

sampling technique for cross-validation. This method maintains the same proportion of classes in both sets, thus avoiding over-representation of any one class in either the training or validation set. The model used the SoftMax activation function to predict the various seagrass categories. To prevent overfitting, we employed the early stopping technique and monitored the validation loss for every iteration on the validation set. The image patch size was set to $400 \times 400$ pixels. We set Patience to 20 to save the best model from the five folds cross-validation training. To make a fair comparison, all the DNEs were trained with a fixed population size of 30, and the number of iterations and the number of runs is set to 20 and 10, respectively. All the DNEs optimised the same nine hyperparameters. To obtain the most suitable parameter values for these models, a trial-and-error approach was used based on a greedy search process.

The proposed model was also trained and tested on the 'DeepSeagrass' dataset with five folds (where the value of K = 5) cross-validation technique. The image patch size was set to $520 \times 578$ (we did not change the original patch size as created in the dataset). The best model was saved from the five-fold cross-validation training and tested on the unseen test set.

We evaluated the proposed model by applying it to the entire raw test images from the 'ECU-MSS-2' dataset. The model classified each patch within these images into four different classes. We further visualized the classification results by assigning a unique colour to each of the four classes for each patch.

To train and test all DNEs, a virtual environment was created using python version 3.7.10. TensorFlow version 2.8.0, and TensorFlow-GPU version 2.4.1 were installed in the environment. All the models were trained and tested using the 'Lambda Blade' server machine (https://lambdalabs.com/deep-learning/servers/blade/basic/customize) which is equipped with 8 NVIDIA GTX 1080 Ti GPUs and has a VRAM capacity of 11GB.

## 2.6 Evaluation metrics

Evaluation metrics are used to measure the performance of deep learning models. The choice of evaluation metric depends on the type of problem and the metric used during training. We evaluated our proposed model using five evaluation metrics: overall accuracy, precision, recall, F1 score and standard deviation. We also visualize the performance using the confusion matrix and convergence curves.

The straightforward and popular performance metric is overall accuracy. Accuracy is the fraction of correctly predicted samples to all the samples and ranges between 0 and 1.

$$Accuracy = \frac{TP + TN}{TP + FP + FN + Tn}$$

Where TP is the number of True Positives (positive samples correctly classified as positive); TN is the number of True negatives (negative samples actually predicted as negative); FP is the number of False positives (negative samples incorrectly detected as positive), and FN is the number of False Negative (the total number of undetected positive samples in the prediction output).

Precision and recall are frequently used in machine learning. Precision defines the proportion of samples that are properly categorised out of all the samples. The recall rate defines the proportion of correctly classified samples out of the total related samples.

$$Precision = \frac{TP}{TP + FP}$$

$$Recall = \frac{TP}{TP + FN}$$

The F1-score is a measure of a model's accuracy that takes both precision and recall into account. It is calculated as the harmonic mean of precision and recall, with a higher score indicating better performance. The F1-score is particularly useful when the distribution of the target variable is imbalanced. The F1-score is normalised between 0 to 1. An F1-score of 1 presents a perfect balance of precision and recall, which are also inversely related too. A confusion matrix is a $N \times N$ table matrix that is used to define the performance of a classification algorithm. Each row of the matrix represents the instances in a predicted class, while each column represents the instances in an actual class (or vice versa). This matrix compares the actual samples with the predicted samples of a model.

## 3. Datasets description

This section describes two multi-species seagrass datasets used in this article, first the 'ECU-MSS-2' dataset and then the publicly available multi-species seagrass dataset.

### 3.1 'ECU-MSS-2' dataset

The 'ECU-MSS-2' dataset contains four different habitats, '*Amphibolis*' spp (hereafter '*Amphibolis*'), '*Halophila*' sp (hereafter '*Halophila*'), '*Posidonia*' spp (hereafter '*Posidonia*') and 'Background'. We compiled this image dataset from different sources. The '*Halophila*' images were collected by the Centre for Marine Ecosystems Research, Edith Cowan University, Western Australia (https://www.ecu.edu.au/schools/science/research-activity/centre-for-marine-ecosystemsresearch/overview). These images were collected from different locations in Western Australia: Fremantle, Mandurah, Rottnest Island, Dampier reserve, Pelican Point, Rocky Bay Swan River, Milyu and Lucky Bay. The images were captured by a Fujifilm X-T30 mirrorless digital camera with wXF 18-55mm F 2.8–4 R LM OIS, all under natural light.

The '*Amphibolis*' 'Background', and '*Posidonia*' images were collected by the Department of Biodiversity, Conservation and Attractions (DBCA), Australia (https://www.dbca.wa.gov.au/) from Shark Bay Marine Park, Shoalwater Island Marine Park, Jurien Bay Marine Park, Ngari Capes Marine Park, Cockburn Sound and the Marmion Marine Park, all in Western Australia. The images were captured by a downward-facing camera held 1- metre above the seagrass canopy with a field of view of 75 × 60 cm along transects used for measuring seagrass density/ height. Ten downward-facing images were collected from each of the three transects per site (30 images per site in total).

Four sample images from the 'ECU-MSS-2' dataset are presented in Fig 4. The '*Amphibolis*' class includes the seagrass species *Amphibolis griffithii* and *Amphibolis antarctica*. The

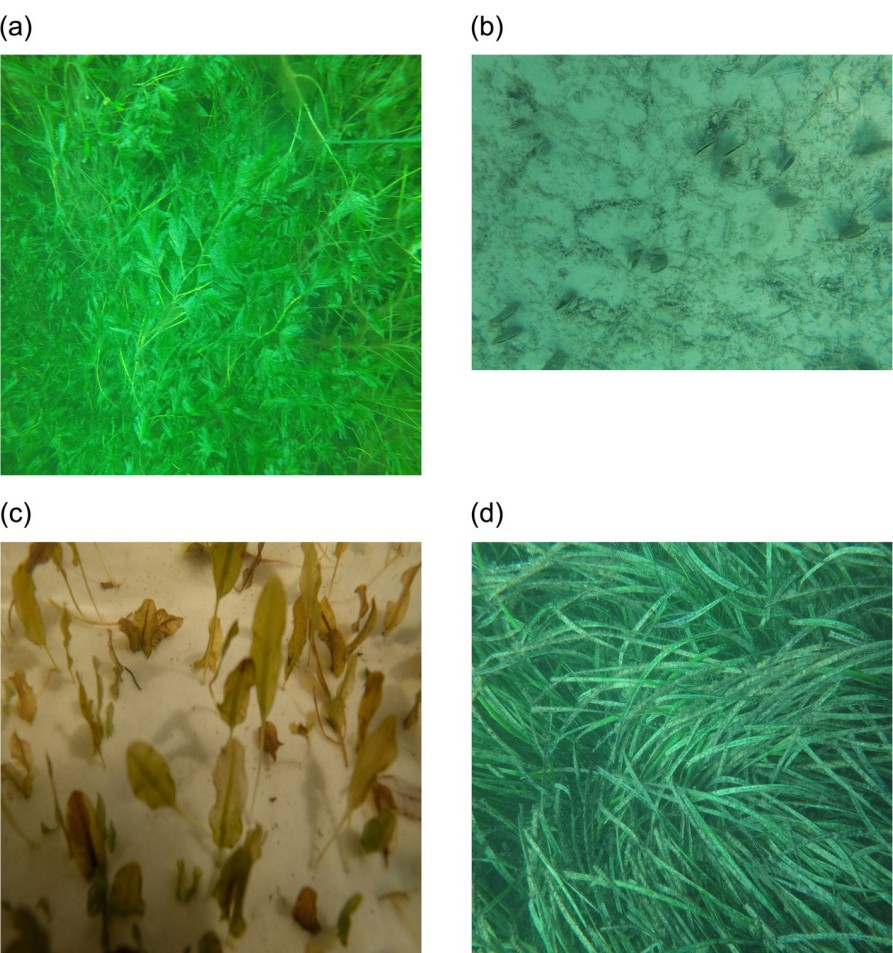

**Fig 4. Sample images from the 'ECU-MSS-2' dataset.**

'*Halophila*' class includes *Halophila ovalis*, while the '*Posidonia*' class includes *Posidonia sinuosa*, *Posidonia coriacea*, and *Posidonia australis* and the 'Background' class includes coral, sand, sponge, seaweeds, fish and other benthic debris.

The total dataset contained 5,201 images, where the '*Amphibolis*' class had 1,304, the 'Background' class had 1,237, the '*Halophila*' class had 1,315 and the '*Posidonia*' class had 1,345 images. The total images were divided into training and test sets. The training set has 4,161 images and the test set had 1,040 images. All four classes had the same 260 images in the test set.

Each image was divided into a grid of 6 rows and 6 columns generating 36 image patches from each image and resulting in a total of 149,796 training and 37,440 test image patches. A CNNs model training on a patch-based dataset requires less processing time and memory than the full image dataset. Once the images were divided into patches, each patch was labelled with the same image label as the seagrass species it represented. Sample patch division is presented in Fig 5. Fig 5a and 5b present two full images, while Fig 5c and 5d present 36 patches generated from the above full images.

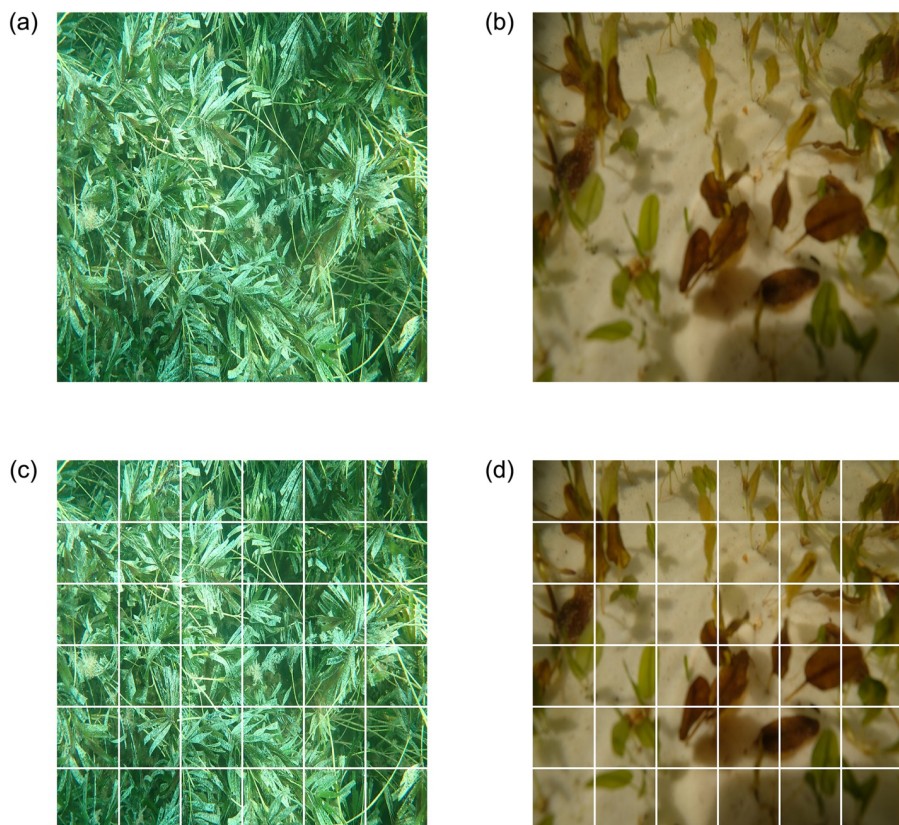

**Fig 5. Sample image patch division performed on the 'ECU-MSS-2' dataset.**

## 3.2 'DeepSeagrass' dataset

We also conducted our experiments on the publicly available multi-species 'DeepSeagrass' [16] dataset. The authors [16] only released image patches, not full images. This dataset contains a total of 66,946 image patches: a training set of 42,848 image patches, a validation set containing 10,720 image patches and a test set containing 13,378 image patches. The dataset has four classes: '*Ferny*' -dense (hereafter '*Ferny*'), '*Strappy*'-dense (hereafter '*Strappy*'), 'Rounded'-dense (hereafter 'Rounded') and 'Background', the first three refer to the general morphology of the seagrass leaves. The images classified as 'Background' were separated into two additional categories: 'Water' and 'Substrate'. The '*Strappy*' class includes *Syringodium isoetifolium*, *Halodule uninervis*, *Cymodocea serrulata* and *Zostera muelleri*, the 'Rounded' class includes *Halophila ovalis* and the '*Ferny*' class includes *Halophila spinulosa* seagrass species.

A Sony Action FDR- 3000X camera was used to capture all of the images from approximately 0.5m distance from the seafloor at around 45 degrees oblique angle. Each image captured image has only one specific seagrass morphotype. All the images were divided into a grid of 5 rows by 8 columns. This patch division generated 40 patches from each image. The patches were 520 × 578 pixels. Due to the low visibility conditions, the top row patches were discarded from each image. Seagrass images were selected with at least 70% seagrass coverage and the background images were selected with less than 1% seagrass coverage. Sample image patches from this dataset are presented in Fig 6.

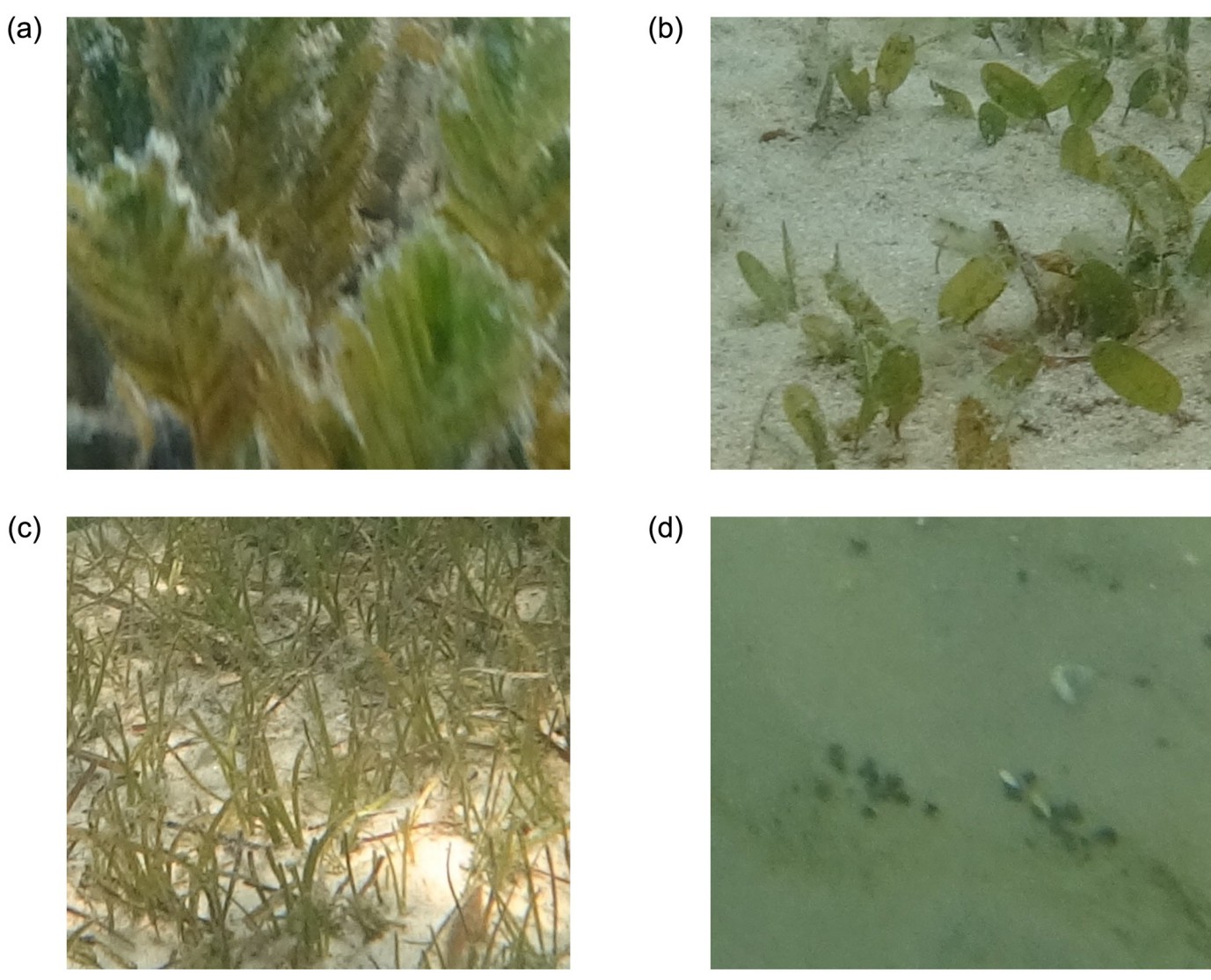

**Fig 6. Sample image patches from the 'DeepSeagrass' dataset.**

## 3.3 Quantitative evaluation

This section is divided into three different subsections. The first subsection describes the performance of all DNEs on the 'ECU-MSS-2' dataset, the second subsection describes the performance of the proposed model on the four classes version of the 'DeepSeagrass' dataset and the third subsection describes the performance of the proposed model on the five classes version of the 'DeepSeagrass' dataset.

**3.3.1 Performance of the proposed algorithm on 'ECU-MSS-2' dataset.** The performance comparison of all the DNEs on the 'ECU-MSS-2' dataset is presented in Table 2. The table presents the best, worst, average overall accuracy and standard deviation (SD) of the 10 different runs. The proposed model achieved the highest best (97.48%), worst (95.14%), average overall accuracy (96.12%) and the lowest (0.0074) standard deviations (SD) (Table 2). The AOS-CNN algorithm achieved the second-highest best overall accuracy (96.94%). SSA-CNN

**Table 2. Quantitative performance of seven evolutionary algorithm-based CNNs on 'ECU-MSS-2' dataset.**

| Model | Best accuracy | Worst accuracy | Average acuracy | Standard Deviation (SD) |
|---|---|---|---|---|
| GWO-CNN [40] | 0.9470 | 0.9145 | 0.9252 | 0.0143 |
| PSO-CNN [40] | 0.9521 | 0.9137 | 0.9395 | 0.0096 |
| CSSA-CNN [40] | 0.9590 | 0.9297 | 0.9465 | 0.0100 |
| SSA-CNN [40] | 0.9617 | 0.9382 | 0.9508 | 0.0077 |
| MFO-CNN [40] | 0.9639 | 0.9480 | 0.9522 | 0.0127 |
| AOS-CNN | 0.9694 | 0.9449 | 0.9556 | 0.0083 |
| **Proposed (BAOS-CNN)** | **0.9748** | **0.9514** | **0.9612** | **0.0074** |

algorithms achieved the second lowest standard deviation (0.0077). GWO-CNN model achieved the lowest (92.52%) and average overall accuracy (92.52%) and the highest standard deviations (0.0143). The lowest standard deviation (0.0074) of the proposed model indicated that the Lévy flight-based AOS algorithm achieved 10 different accuracies that were very close to the average accuracy (96.12%). Accuracies obtained by the proposed algorithm were clustered in the small range [97.48% to 95.14%] indicating that the proposed algorithm optimised the CNNs more accurately.

The detailed performance of these DNEs for four classes: 'Amphibolis', 'Background', 'Halophila', and 'Posidonia' on the 'ECU-MSS-2' dataset is presented in Table 3. The proposed algorithm achieved the highest overall accuracy (97.48%) among all the DNEs. For the 'Amphibolis' class, the proposed model achieved the highest F1-score (94.98%), the CSSA-CNN achieved the highest precision (98.25%) and the SSA-CNN achieved the highest recall (92.88%). For the 'Background' class, the proposed model achieved the highest F1-score (99.16%), the AOS-CNN achieved the highest precision (99.46%), and the CSSA-CNN achieved the highest recall (99.24%).

For the '*Halophila*' class, the proposed model achieved the highest F1-score (99.96%) with the second highest precision (99.98%) and recall (99.94%). For the '*Posidonia*' class, the proposed algorithm achieved the highest precision (93.54%)and the highest F1-score (95.80%). Overall, the table also indicates that the AOS-CNN algorithm achieved the second-highest overall accuracy (96.94%) and second-highest F1-scores of all four classes. The AOS-CNN algorithm achieved the highest precision (99.46%) for 'Background' and the highest recall (99.98%) for the '*Halophila*' class.

The confusion matrix of the proposed BAOS-CNN model is presented in Fig 7. A confusion matrix can be formed from the predicted and actual values. Note that the higher values of true

**Table 3. The detailed performance of the best models on the 'ECU-MSS-2' dataset.**

| Model | *Amphibolis* | | | Background | | | *Halophila* | | | *Posidonia* | | | Overall accuracy |
|---|---|---|---|---|---|---|---|---|---|---|---|---|---|
| | Precision | Recall | F1-score | Precision | Recall | F1-score | Precision | Recall | F1-score | Precision | Recall | F1-score | |
| GWO-CNN [40] | 0.9659 | 0.8592 | 0.9094 | 0.9944 | 0.9436 | 0.9683 | **0.9999** | 0.9953 | 0.9976 | 0.8490 | **0.9901** | 0.9141 | 0.9470 |
| PSO-CNN [40] | 0.9812 | 0.8288 | 0.8986 | 0.9868 | 0.9913 | 0.9891 | 0.9987 | 0.9990 | 0.9989 | 0.8600 | 0.9893 | 0.9202 | 0.9521 |
| CSSA-CNN [40] | **0.9825** | 0.8572 | 0.9156 | 0.9831 | **0.9924** | 0.9877 | 0.9995 | 0.9974 | 0.9984 | 0.8829 | 0.9889 | 0.9329 | 0.9590 |
| SSA-CNN [40] | 0.9221 | **0.9288** | 0.9255 | 0.9930 | 0.9754 | 0.9842 | 0.9995 | 0.9973 | 0.9984 | 0.9333 | 0.9451 | 0.9392 | 0.9617 |
| MFO-CNN [40] | 0.9714 | 0.8881 | 0.9279 | 0.9828 | 0.9912 | 0.9870 | 0.9982 | 0.9978 | 0.9980 | 0.9082 | 0.9786 | 0.9421 | 0.9639 |
| AOS-CNN | 0.9778 | 0.9053 | 0.9402 | **0.9946** | 0.9838 | 0.9892 | 0.9985 | **0.9998** | 0.9991 | 0.9125 | 0.9889 | 0.9491 | 0.9694 |
| **Proposed (BAOS-CNN)** | 0.9732 | 0.9275 | **0.9498** | 0.9926 | 0.9905 | **0.9916** | 0.9998 | 0.9994 | **0.9996** | **0.9354** | 0.9817 | **0.9580** | **0.9748** |

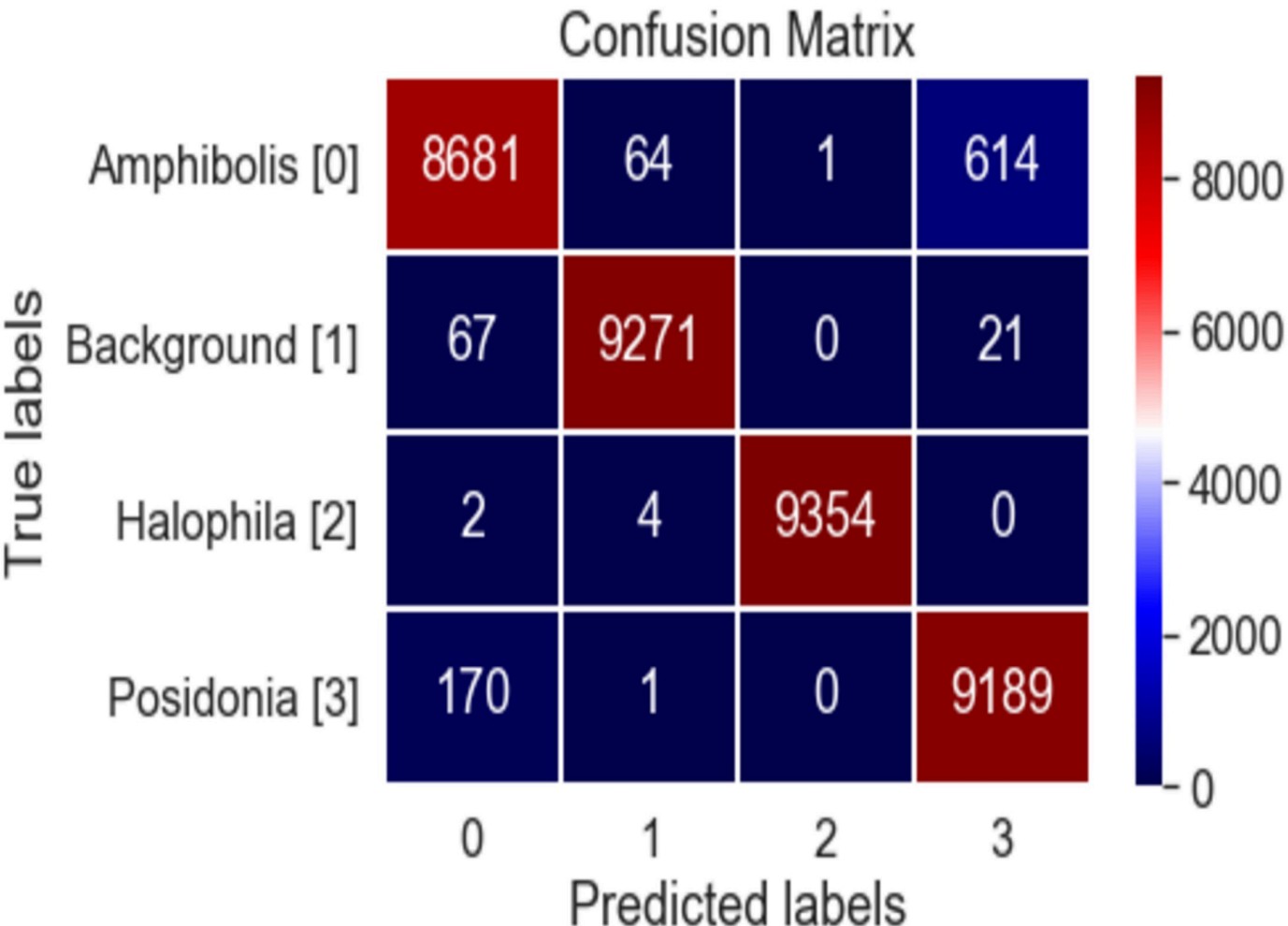

**Fig 7. Confusion metrics of the proposed OFDA-CNN algorithm.**

positive and true negative indicate the higher accuracy of the classifier. The accuracy metric can be misleading when the dataset is imbalanced, and the confusion matrix can, therefore, help assess performance. The proposed BAOS-CNN classifier correctly classified 8681 of 9360 '*Amphibolis*' image patches, while 614 '*Amphibolis*' image patches were misclassified as '*Posidonia*' images. The proposed classifier misclassified 614 '*Amphibolis*' patches due to their strap-like shape appearance in some image patches. This strap-like shape has a similar look to '*Posidonia*'. The proposed algorithm classified 9271 'Background' image patches correctly, with 67 'Background' image patches were misclassified as '*Amphibolis*'. The proposed classifier detected most of the '*Halophila*' image patches (9354) correctly, with only 6 misclassified image patches. The proposed classifier detected 9189 '*Posidonia*' image patches correctly, with 171 misclassifications.

**3.3.2 Performance of the proposed algorithm on publicly available 'DeepSeagrass' dataset (four classes version).** The performance of the proposed algorithm compared to two previously reported classifiers [16, 17] on the 'DeepSeagrass' dataset is presented in Table 4. The four classes version of the 'DeepSeagrass' dataset has '*Strappy*', '*Ferny*', 'Rounded' and 'Background' classes. The proposed algorithm achieved the highest overall accuracy (92.30%) and

**Table 4. Performance of the proposed model on four classes version of the 'DeepSeagrass' dataset.**

| Model | *Strappy* | | *Ferny* | | *Rounded* | | Background | | Overall Accuracy |
|---|---|---|---|---|---|---|---|---|---|
| | Precision | Recall | Precision | Recall | Precision | Recall | Precision | Recall | |
| Raine et al. [16] | 0.836 | **0.954** | 0.842 | 0.964 | 0.862 | 0.86 | **0.976** | 0.774 | 0.882 |
| Noman et al. [17] | 0.920 | 0.950 | 0.860 | **0.970** | 0.890 | 0.890 | 0.960 | **0.830** | 0.913 |
| Proposed (BAOS-CNN) | **0.931** | 0.945 | **0.882** | 0.945 | **0.944** | **0.953** | 0.963 | 0.812 | **0.923** |

**Table 5. Performance of proposed model on five classes version of the 'DeepSeagrass' dataset.**

| Model | *Strappy* | | *Ferny* | | *Rounded* | | Substrate | | Water | | Overall Accuracy |
|---|---|---|---|---|---|---|---|---|---|---|---|
| | Precision | Recall | Precision | Recall | Precision | Recall | Precision | Recall | Precision | Recall | |
| Raine et al. [16] | 0.892 | 0.952 | 0.938 | **0.960** | 0.898 | 0.858 | 0.94 | 0.982 | 0.982 | 0.258 | 0.924 |
| Noman et al. [17] | 0.850 | **0.980** | 0.960 | 0.940 | 0.910 | 0.910 | 0.950 | 0.990 | **0.990** | 0.150 | 0.930 |
| Proposed (BAOS-CNN) | **0.939** | 0.976 | **0.968** | 0.850 | **0.994** | **0.998** | **0.993** | **0.993** | 0.997 | **0.407** | **0.935** |

outperformed previous state-of-the-art accuracy (91.30%) reported by Noman et al. [17]. While the proposed algorithm delivered the highest overall accuracy, this was not always the case for the individual image classes. For the '*Strappy*' class, the proposed algorithm achieved the highest precision (93.1%) and Raine et al. [16] achieved the highest recall (95.4%). For the '*Ferny*' class, the proposed algorithm achieved the highest precision (88.2%) and Noman et al. [17] achieved the highest recall (97.0%). For the 'Rounded class, the proposed algorithm achieved the highest precision (94.4%) and highest recall (95.3%). For the 'Background' class, Raine et al. [16] achieved the highest precision (97.6%) and Noman et al. [17] the highest recall (83.0%).

**3.3.3 Performance of the proposed algorithm on publicly available 'DeepSeagrass' dataset (five classes version).** The performance of the proposed algorithm with two previously reported classifiers on the five classes version of the 'DeepSeagrass' dataset is presented in Table 5. The five classes version of the 'DeepSeagrass' dataset has '*Strappy*', '*Ferny*', 'Rounded', 'Water' and 'Substrate' classes. The proposed model achieved the highest 93.5% overall accuracy, which is higher than the previously reported state-of-the-art overall accuracy of 93.0 (reported by Noman et al. [17]). The proposed model achieved the highest precision and recall (around 99%) in the 'Rounded' and 'Substrate' classes. In the '*Strappy*' class, the proposed model achieved the highest precision (93.9%) and the second highest recall (97.6%), whereas the Noman et al. (EfficientNet-based classifier) [17] the highest recall (98.0%). In the '*Ferny*' class, the proposed classifier achieved the highest precision (96.8%), whereas Raine et al. (VGG16-based classifier) [16] achieved the highest recall (96.0%). In the 'Water' class, the proposed classifier achieved the highest recall (40.7%) and second highest precision (99.7%), whereas Noman et al. [17] achieved the highest accuracy (99.0%).

### 3.4 Qualitative evaluation

The proposed algorithm was applied to label (print) the class name in the image patches automatically. Fig 8a shows that the proposed algorithm predicted '*Amphibolis*' as '*Amphibolis*' correctly, although this '*Amphibolis*' image patch contains stems of '*Amphibolis*', which have a very similar shape to '*Posidonia*'. Fig 8b shows that the model predicted 'Background' correctly, although this patch contains seaweeds, which have a similar shape and colour to '*Halophila*'. Fig 8c contains a very small amount of '*Halophila*', although, the proposed model

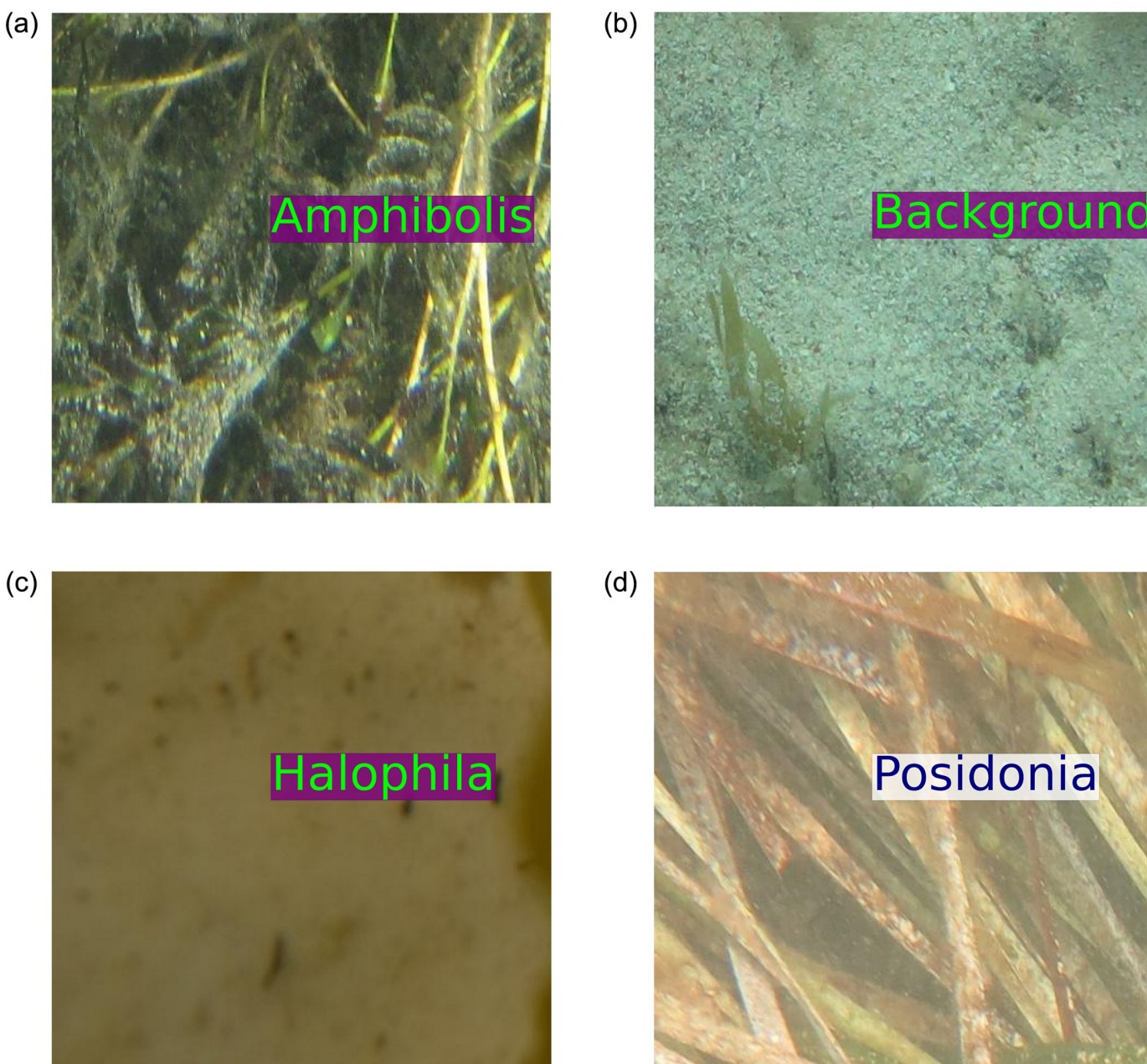

**Fig 8. Sample classified images from the 'ECU-MSS-2' dataset.** Label inside the image patches is predicted by the proposed model.

detected this successfully. Fig 8d presents some dead '*Posidonia*' seagrass. Surprisingly, the proposed model labelled the patch correctly. Sample misclassifications from the 'ECU-MSS-2' dataset are presented in Fig 9.

Fig 9a and 9d present misclassifications of '*Amphibolis*' as '*Posidonia*'. The proposed model was misguided by the stem of the '*Amphibolis*' seagrass, which has a similar, linear, shape to '*Posidonia*'. Fig 9b represents the misclassification'Background' as '*Amphibolis*' and contains macroalgae (seaweed), which misguided the proposed model. Fig 9c shows the misclassification of 'Background' as '*Halophila*', again likely due to the presence of macroalgae. Fig 10

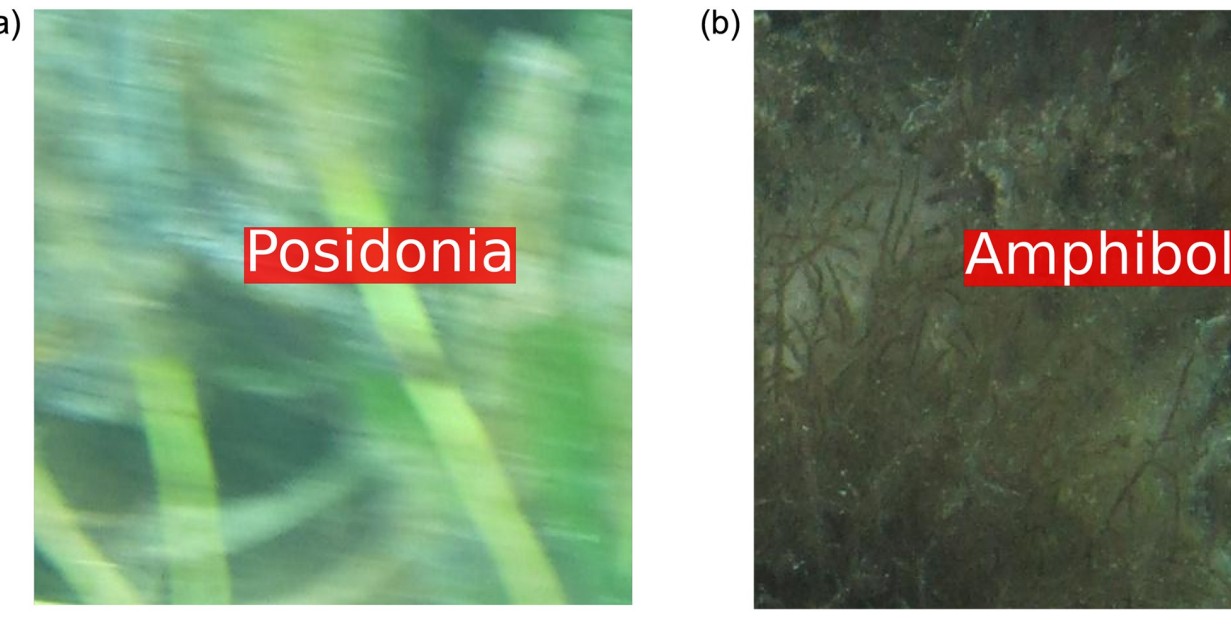

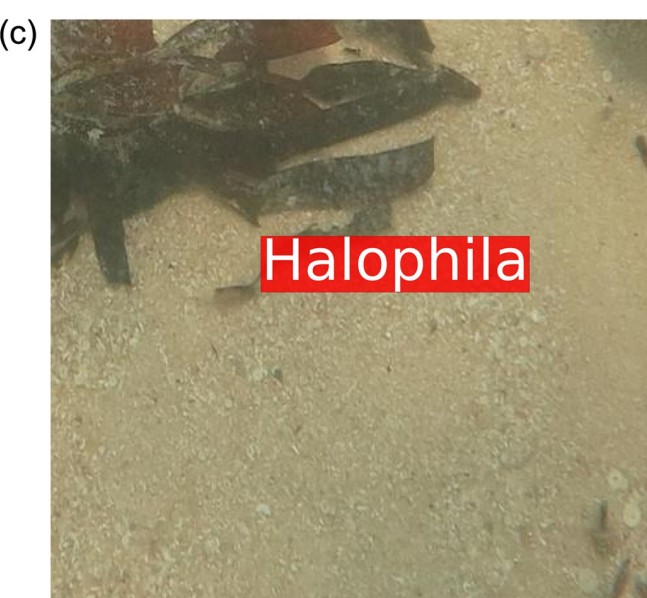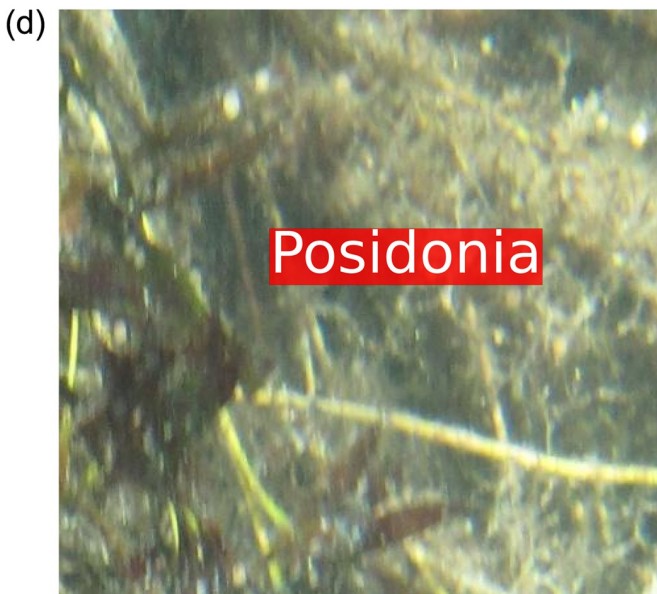

**Fig 9. Sample misclassified images from the 'ECU-MSS-2' dataset.** Label inside the image patches is predicte by the proposed model.

demonstrates the sample correct classification of the proposed model from the 'DeepSeagrass' data. The figure also demonstrates that all the seagrass image patches '*Strappy*', '*Ferny*', and 'Rounded' are very hard to identify for human eyes. However, the proposed model correctly classified all four classes of image patches and labelled the class names on the images automatically.

Fig 11 presents sample misclassifications of image patches from the 'DeepSeagrass' dataset. The figure demonstrates that the proposed model predicted the class name and labelled the class names on the image patches automatically. The ground truths are presented in the

(a) 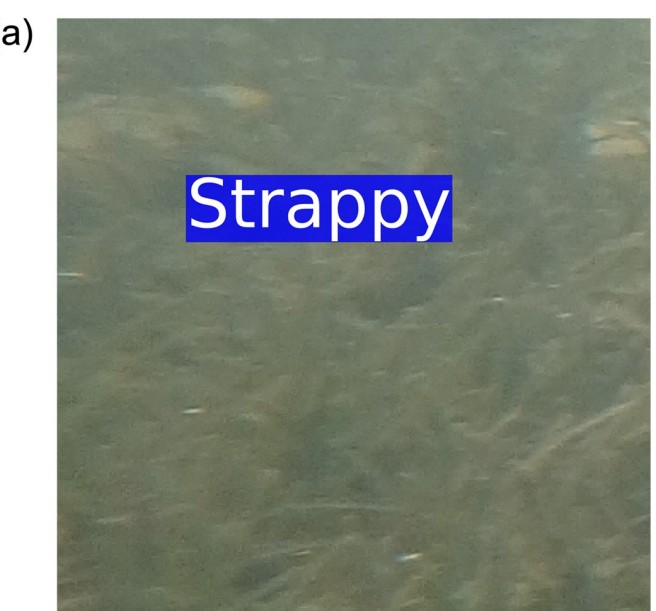

(b) 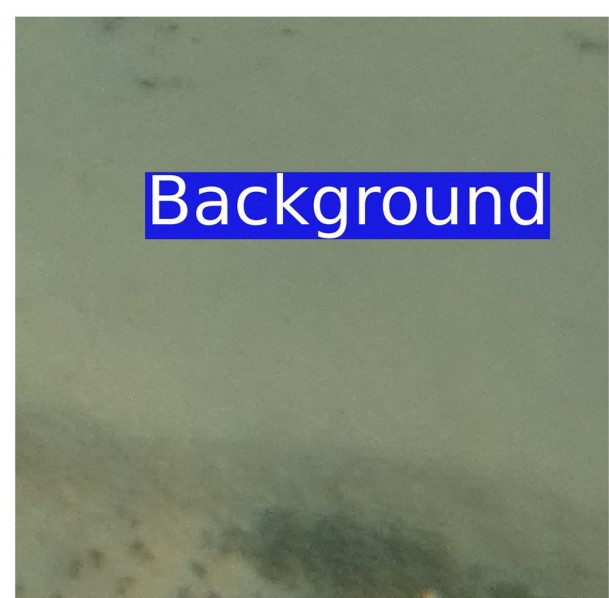

(c) 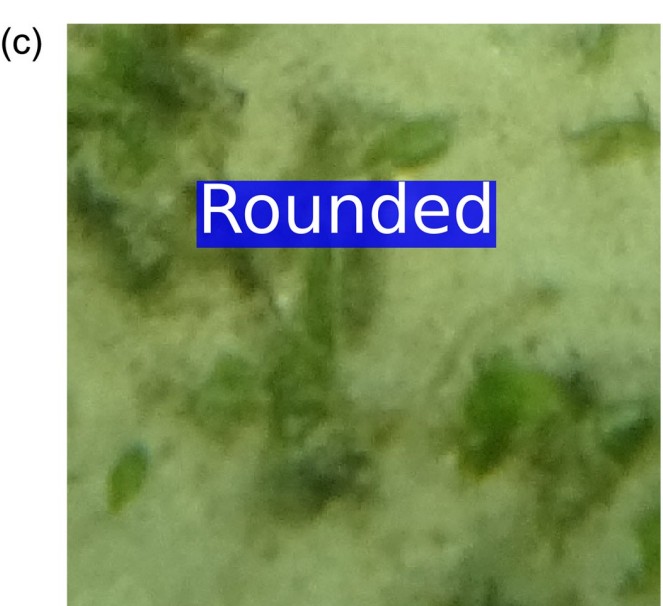

(d) 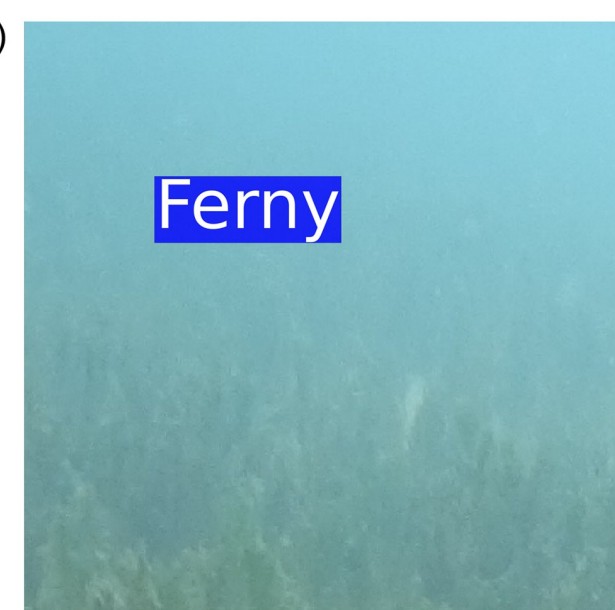

**Fig 10. Sample correctly classified image patches from the 'DeepSeagrass' dataset.** Label inside the image patches is predicted by the proposed model.

figures' captions. Fig 11a presents that the proposed model misclassified the 'Background' image patch as '*Strappy*'. This image patch contains '*Strappy*' seagrass. Due to the patch division, the patch was incorrectly labelled as 'Background' in the ground truth. However, the proposed model predicted the class correctly. Fig 11b demonstrates the misclassification of the '*Ferny*' class as 'Background'. The proposed model was misguided with the underwater environment. The image patch was captured with a long-distance view and the seagrass is not clear to distinguish. Fig 11c represents the misclassification of the 'Rounded' image patch as

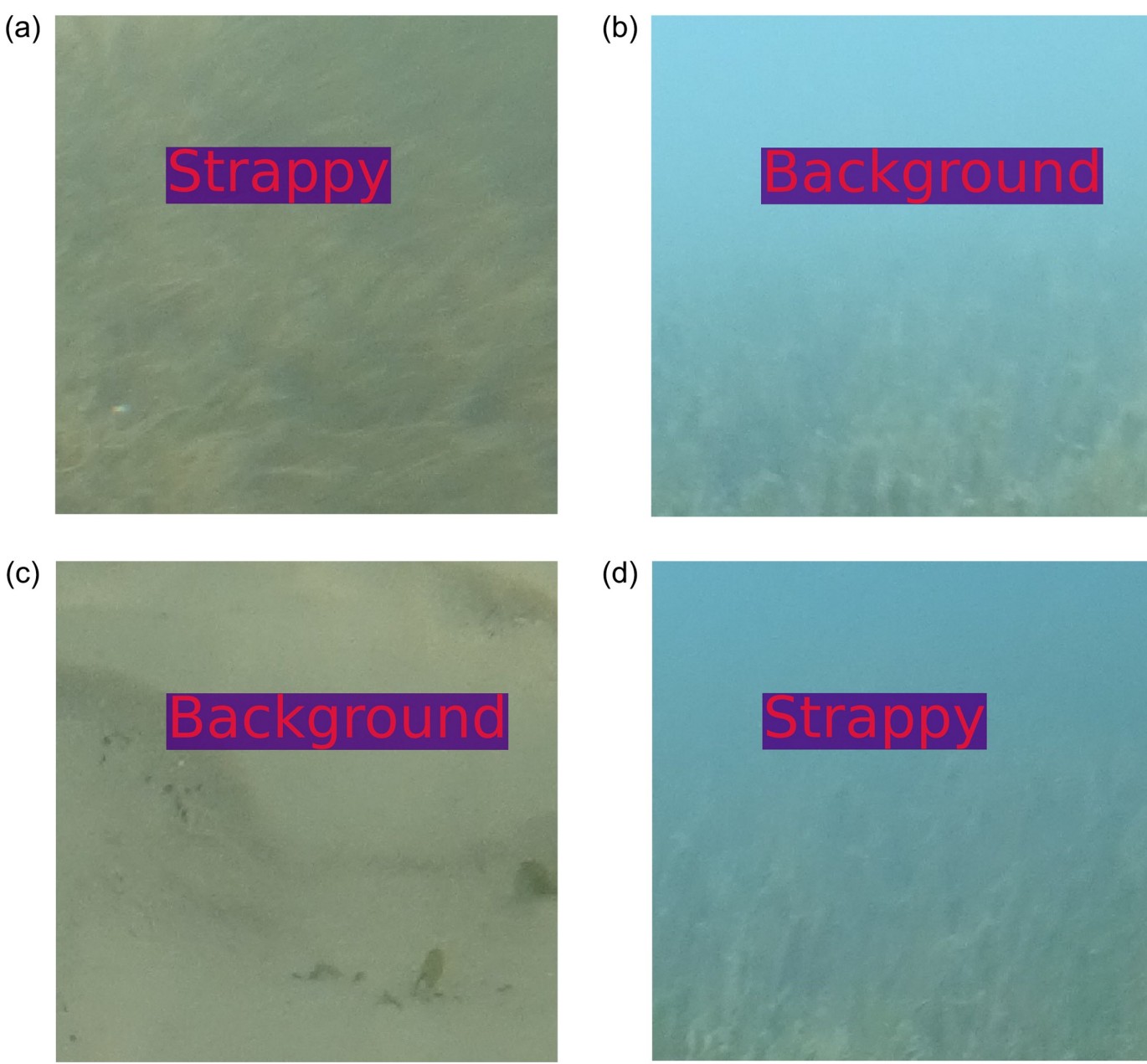

**Fig 11. Sample misclassified image patches from the 'DeepSeagrass' dataset.** Label inside the image patches is predicted by the proposed model.

'Background'. The image patch contains only one small *Halophila ovalis* seagrass, which was missed by the model. Fig 11d represents the '*Ferny*' class misclassified as '*Strappy*'. The proposed model was misguided with the blurriness and underwater environment.

The proposed model was also applied to detect patches on the whole test images intofour classes: '*Amphibolis*', 'Background', '*Halophila*' and '*Posidonia*'. The test images were overlaid with four different colors representing the class labels. Patches were classified correctly in all of the image patches, as illustrated in Fig 12. Although Fig 12c image contains two different classes of image patches such as '*Halophila*' and 'Background', the proposed model detected both

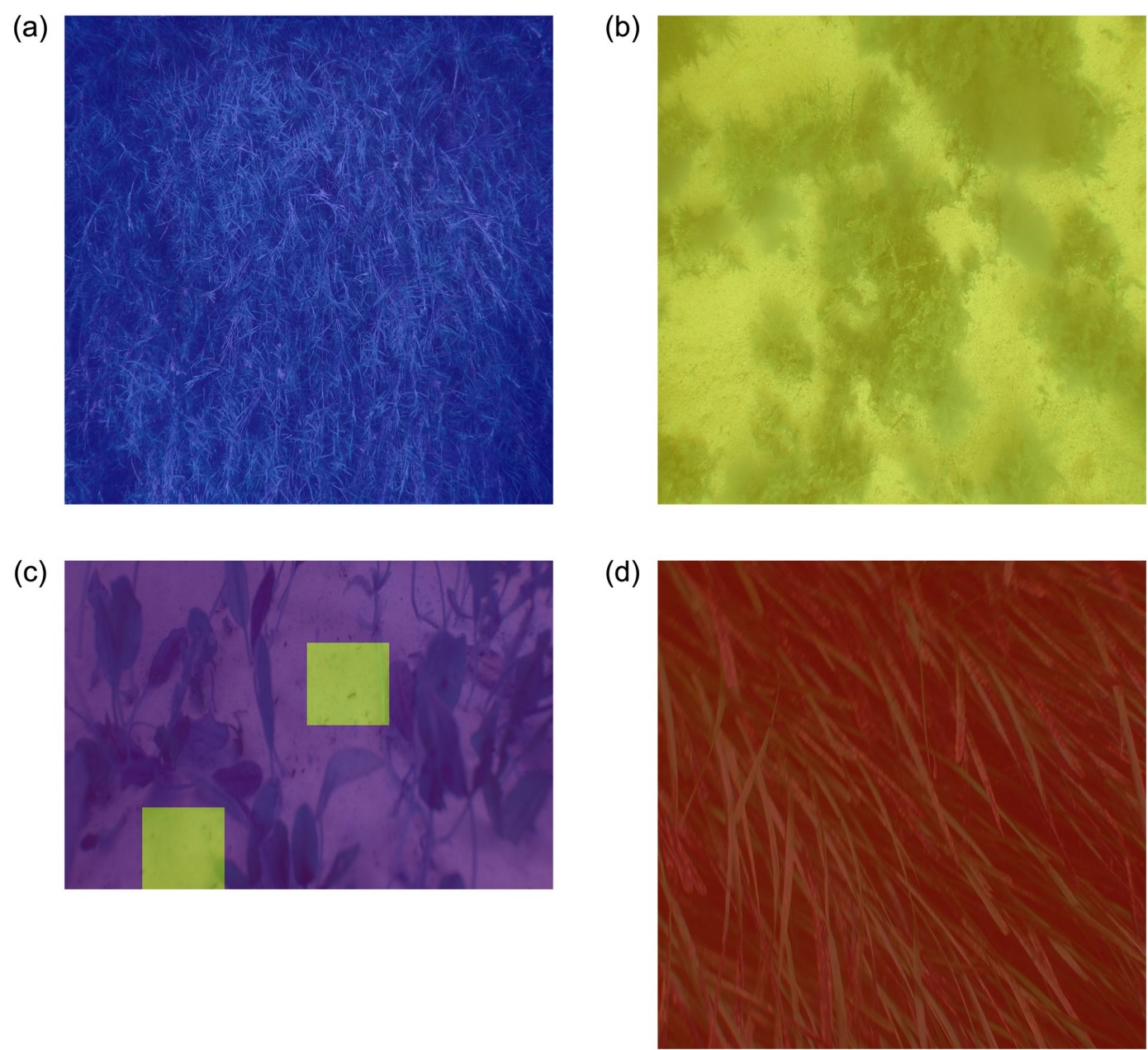

**Fig 12. Test images with classified patches (where blue presents '*Amphibolis*', yellow presents 'Background', purple presents '*Halophila*' and maroon presents '*Posidonia*').**

classes correctly. This visualization can be useful for seagrass mapping applications. Fig 13 demonstrates some misclassification of image patches. The proposed model misclassified some '*Amphibolis*' image patches as '*Posidonia*' patches. This model misclassified these patches due to minimal inter-class differences and the stem of '*Amphibolis*' having a similar shape to '*Posidonia*'.

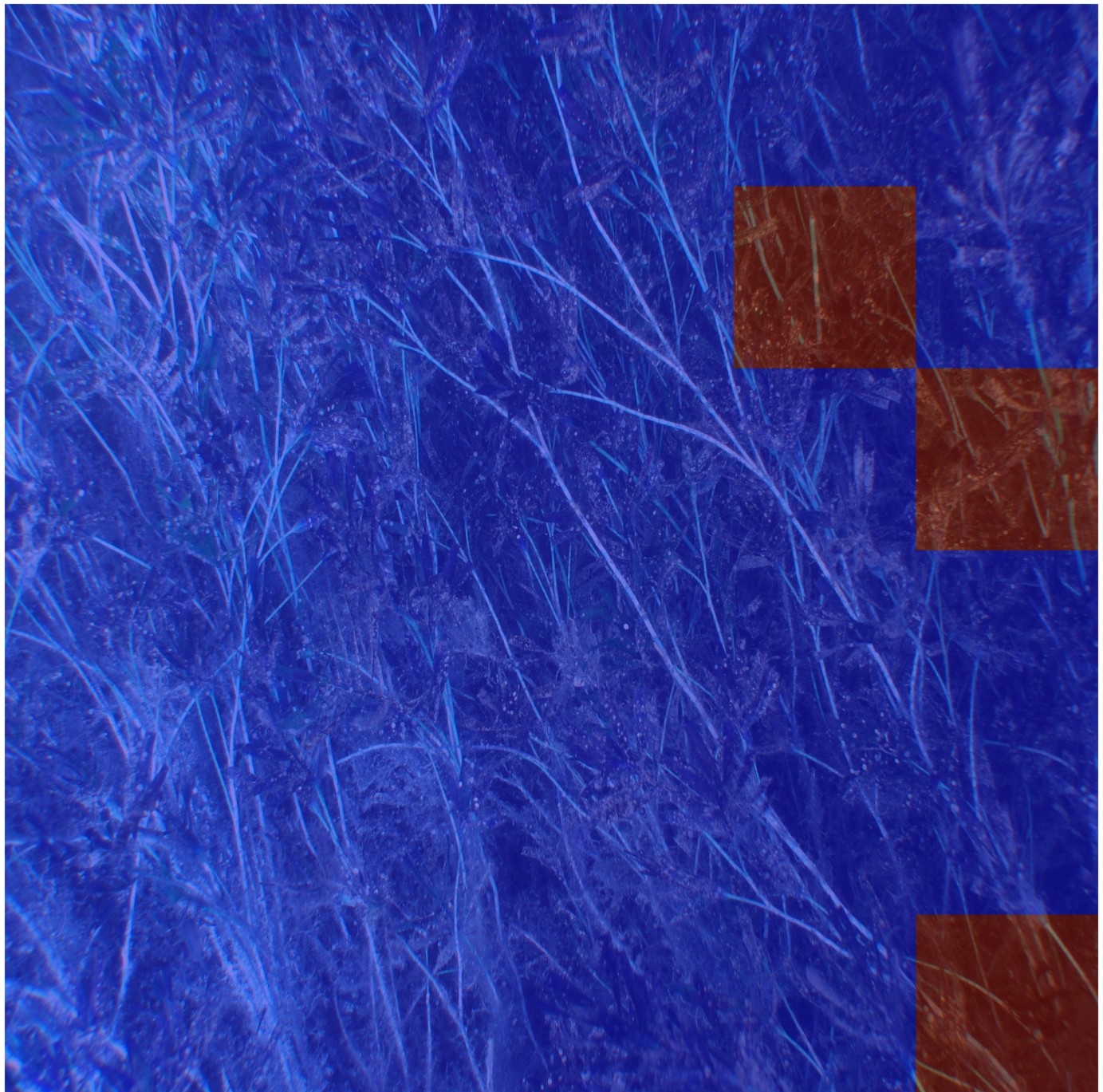

**Fig 13. Test images with misclassified patches (where blue presents '*Amphibolis*' and maroon represents '*Posidonia*').**

### 3.5 Convergence curve

The convergence curves of the different DNEs based on accuracy are shown in Fig 14. The figure visualizes the exploration ability of different DNEs in finding the optimal solution at high speed. The proposed model started leading among all the DNEs from the first epoch. After six

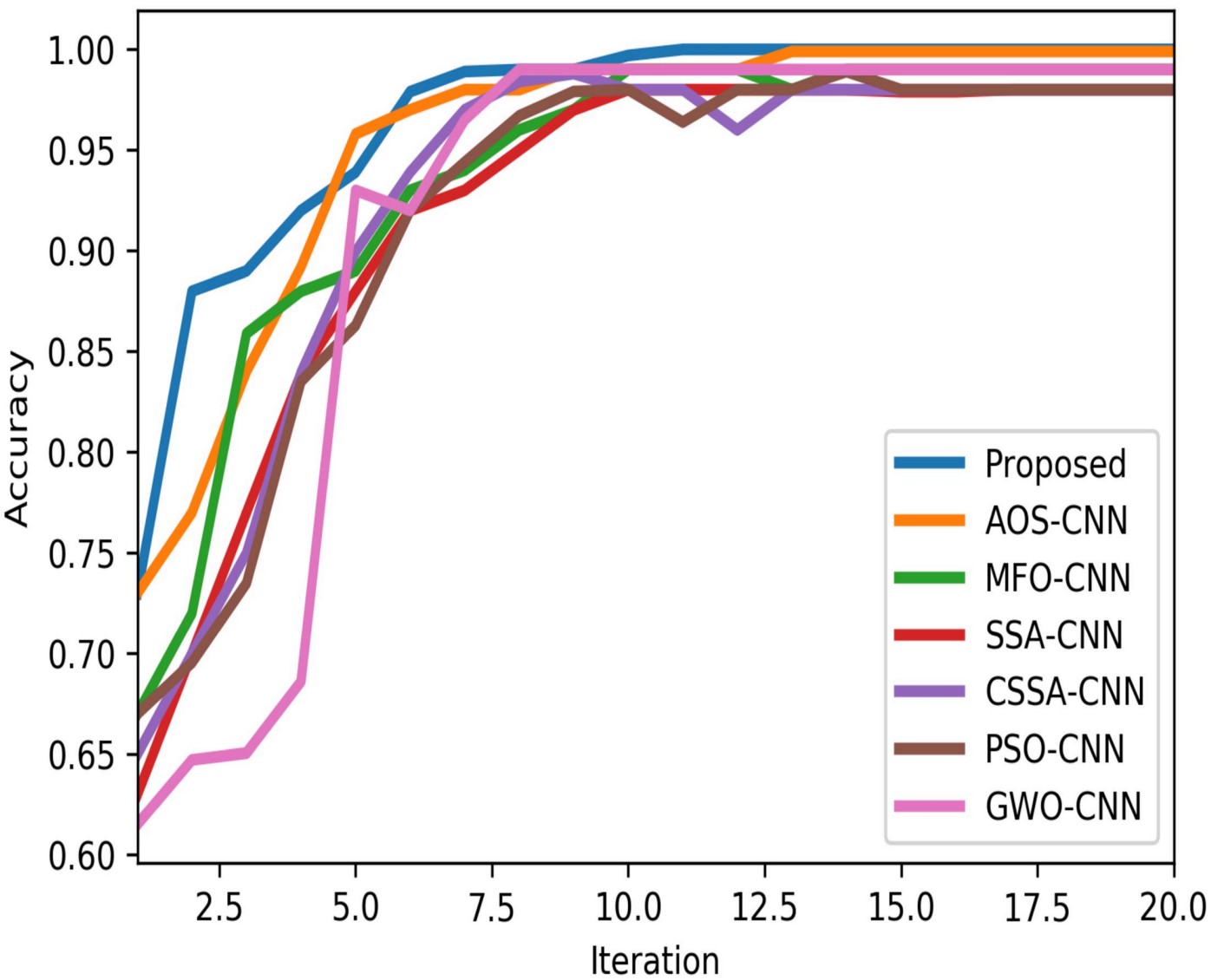

**Fig 14. Convergence curves of different DNEs based on the accuracy metric.**

or seven epochs, it again led among all the curves. The figure also shows that all models have started with 64–74% accuracy and reached almost 97–99% accuracy after 6th iterations. The convergence curve of the proposed model started with around 74% accuracy and ended with 99%.

## 4. Conclusion

This paper presents the first development of a novel metaheuristic algorithm (BAOS) by combining the Lévy flight technique with the recently proposed Atomic Orbital Search algorithm. It also presents a novel deep neuroevolutionary algorithm (BAOS-CNN) to optimise a CNNs model on a patch-based multi-species seagrass dataset. The proposed BAOS-CNN framework leveraged the power of BAOS to automate the architectural engineering and hyperparameter

tuning of CNNs. The proposed BAOS-CNN achieved 97.48% accuracy, the highest among seven evolutionary algorithms, by creating a random population and evolving and creating a new generation to search the best architecture and hyperparameters combination of the CNNs models on the seagrass dataset. The proposed BAOS-CNN model achieved overall accuracies of 93.50% and 92.30% on the five-classes version and four-classes version of the publicly available 'DeepSeagrass' dataset, respectively, outperforming the previously reported state-of-the-art accuracies on this dataset. The proposed algorithm can minimize the requirement of manual CNNs model architectural engineering and hyperparameter tuning. The proposed algorithm has great potential for application in image classification or object detection in marine, or other submerged environments, including the development of automated detection and monitoring systems. DNE is a powerful optimization method and an active area of research and new developments are expected in the future. In future, we will apply ensemble learning techniques to enhance the performance by combining the multiple evolutionary algorithm-based CNNs' performances.

## Acknowledgments

The authors would like to acknowledge the School of Science at Edith Cowan University (ECU), WA, for their support. The authors would like to thank Centre for Marine Ecosystems Research (CMER) and Department of Biodiversity, Conservation and Attractions (DBCA) for providing the seagrass images used in this research.

## Author Contributions

**Conceptualization:** Md Kislu Noman, Syed Mohammed Shamsul Islam, Seyed Mohammad Jafar Jalali.

**Data curation:** Md Kislu Noman, Paul Lavery.

**Formal analysis:** Md Kislu Noman, Syed Mohammed Shamsul Islam, Seyed Mohammad Jafar Jalali, Paul Lavery.

**Funding acquisition:** Syed Mohammed Shamsul Islam, Seyed Mohammad Jafar Jalali.

**Investigation:** Syed Mohammed Shamsul Islam, Paul Lavery.

**Methodology:** Md Kislu Noman, Seyed Mohammad Jafar Jalali.

**Project administration:** Syed Mohammed Shamsul Islam, Paul Lavery.

**Resources:** Seyed Mohammad Jafar Jalali, Paul Lavery.

**Software:** Md Kislu Noman.

**Supervision:** Syed Mohammed Shamsul Islam, Jumana Abu-Khalaf, Paul Lavery.

**Validation:** Md Kislu Noman, Jumana Abu-Khalaf, Paul Lavery.

**Visualization:** Md Kislu Noman.

**Writing – original draft:** Md Kislu Noman, Jumana Abu-Khalaf.

**Writing – review & editing:** Md Kislu Noman.

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
