## [Decision Letter · Decision Letter 0]

5 Jun 2023

PONE-D-23-00889BAOS-CNN: A novel deep neuroevolution algorithm for multispecies seagrass detectionPLOS ONE

Dear Dr. Noman,

Thank you for submitting your manuscript to PLOS ONE. After careful consideration, we feel that it has merit but does not fully meet PLOS ONE’s publication criteria as it currently stands. Therefore, we invite you to submit a revised version of the manuscript that addresses the points raised during the review process, and provide response to their suggestions.

We look forward to receiving your revised manuscript.

Kind regards,

Jin Liu

Academic Editor

PLOS ONE

Journal Requirements:

2. Please note that PLOS ONE has specific guidelines on code sharing for submissions in which author-generated code underpins the findings in the manuscript. In these cases, all author-generated code must be made available without restrictions upon publication of the work. 

Please review our guidelines at https://journals.plos.org/plosone/s/materials-and-software-sharing#loc-sharing-code and ensure that your code is shared in a way that follows best practice and facilitates reproducibility and reuse.

"This research work is funded by the Australian Government Research Training Program (RTP) scholarship and supported by School of Science and Centre for Marine Ecosystems Research, Edith Cowan University, Australia.."

"NO - Include this sentence at the end of your statement: The funders had no role in study design, data collection and analysis, decision to publish, or preparation of the manuscript."

7. Please upload a new copy of Figure 5 as the detail is not clear. Please follow the link for more information:

https://blogs.plos.org/plos/2019/06/looking-good-tips-for-creating-your-plos-figures-graphics/

https://blogs.plos.org/plos/2019/06/looking-good-tips-for-creating-your-plos-figures-graphics/

**Additional Editor Comments:**

Based on the advice received, we invite you to submit a revised version of the manuscript that addresses the points raised during the review process.

Reviewers' comments:

Reviewer's Responses to Questions

**Comments to the Author**

1. Is the manuscript technically sound, and do the data support the conclusions?

Reviewer #1: Yes

Reviewer #2: Yes

2. Has the statistical analysis been performed appropriately and rigorously? 

Reviewer #1: Yes

Reviewer #2: Yes

3. Have the authors made all data underlying the findings in their manuscript fully available?

Reviewer #1: Yes

Reviewer #2: Yes

4. Is the manuscript presented in an intelligible fashion and written in standard English?

Reviewer #1: Yes

Reviewer #2: Yes

5. Review Comments to the Author

Reviewer #1: The paper presents Deep Neuroevolutionary (DNE) model

that can automate the architectural engineering

Since the authors are proposing novel deep neuroevolution algorithm, I am not able to find related work in the paper that can justify this novel contribution.

Author must add related work and compare their novel deep neuroevolution algorithm with other similar or near-to-similar proposed algorithms.

How the evaluation of the fitness value of the candidates are computed. Kindly explain flow chart with the help of example

There is no appropriate explanation of algorithm 1 in the paper.

Is ECU-MSS-2 confidential dataset?

Present the comparison between qualitative and quantitative evaluation in tabular format

Reviewer #2: The paper presents a significant advancement in the field of metaheuristic algorithms, by combining the Lévy flight technique with the recently proposed Atomic Orbital Search (BAOS) algorithm. The innovative approach and promising results of the BAOS-CNN framework, particularly in automating architectural engineering and hyperparameter tuning of CNNs, is commendable.

The highlight of the paper is undoubtedly the application of BAOS-CNN to optimize a CNN model on a multi-species seagrass dataset. The achieved accuracy of 97.48%, which outperforms seven other evolutionary algorithms, is impressive. The use of BAOS to create a random population and evolve a new generation for searching the optimal architecture and hyperparameters combination of the CNN models showcases the power and versatility of this novel algorithm.

The authors' success in achieving high accuracy on different versions of the publicly available 'DeepSeagrass' dataset further underscores the effectiveness of the proposed BAOS-CNN model. The fact that it outperforms previously reported state-of-the-art accuracies on this dataset is a substantial accomplishment.

The paper also provides a valuable contribution by minimizing the requirement of manual CNN model architectural engineering and hyperparameter tuning. This feature enhances the practicality and usability of the algorithm in real-world applications, making it a potentially powerful tool in image classification or object detection in marine or other submerged environments.

In terms of future work, the authors' plan to apply ensemble learning techniques to enhance the performance by combining multiple evolutionary algorithm-based CNNs’ performances is promising. This approach could further improve the accuracy of the model and broaden the range of its applications.

Overall, the authors should be commended for their innovative work in developing and applying the BAOS-CNN framework. The paper is well-structured and the findings are clearly presented. The proposed algorithm has great potential and sets the stage for exciting future developments in the field of deep neuroevolutionary algorithms. I look forward to seeing these future developments and the potential real-world applications that this algorithm could enable.

6. PLOS authors have the option to publish the peer review history of their article (what does this mean?). If published, this will include your full peer review and any attached files.

Reviewer #1: No

Reviewer #2: **Yes: **Michał Jasiński

<quillbot-extension-portal></quillbot-extension-portal><quillbot-extension-portal></quillbot-extension-portal>

<quillbot-extension-portal></quillbot-extension-portal><quillbot-extension-portal></quillbot-extension-portal>

---

## [Author Response · Author response to Decision Letter 0]

9 Oct 2023

Dear Reviewers,

We are delighted that you found our work to be a significant advancement in the field of metaheuristic algorithms. Your insightful comments and positive feedback have truly encouraged and motivated us. We have carefully addressed all the comments in our revised manuscript. Thank you so much for your comments.

---

## [Decision Letter · Decision Letter 1]

6 Nov 2023

BAOS-CNN: A novel deep neuroevolution algorithm for multispecies seagrass detection

PONE-D-23-00889R1

Dear Dr. Noman,

We’re pleased to inform you that your manuscript has been judged scientifically suitable for publication and will be formally accepted for publication once it meets all outstanding technical requirements.

Kind regards,

Jin Liu

Academic Editor

PLOS ONE

Additional Editor Comments (optional):

After carring out the reviewers suggestions, this manuscript can be accepted for publication now.

Reviewers' comments:

Reviewer's Responses to Questions

**Comments to the Author**

1. If the authors have adequately addressed your comments raised in a previous round of review and you feel that this manuscript is now acceptable for publication, you may indicate that here to bypass the “Comments to the Author” section, enter your conflict of interest statement in the “Confidential to Editor” section, and submit your "Accept" recommendation.

Reviewer #1: (No Response)

Reviewer #2: All comments have been addressed

2. Is the manuscript technically sound, and do the data support the conclusions?

Reviewer #1: Yes

Reviewer #2: Yes

3. Has the statistical analysis been performed appropriately and rigorously? 

Reviewer #1: Yes

Reviewer #2: Yes

4. Have the authors made all data underlying the findings in their manuscript fully available?

Reviewer #1: Yes

Reviewer #2: Yes

5. Is the manuscript presented in an intelligible fashion and written in standard English?

Reviewer #1: Yes

Reviewer #2: Yes

6. Review Comments to the Author

Reviewer #1: I am not able to find related work in the paper that can

justify this novel contribution

Author must add realted work section with clear contribution

Reviewer #2: I accept revision. no more changes are reqired. So I hope this paper can be accepted now in the PLOS ONE journal.

7. PLOS authors have the option to publish the peer review history of their article (what does this mean?). If published, this will include your full peer review and any attached files.

Reviewer #1: **Yes: **Asadullah Shaikh

Reviewer #2: **Yes: **Michał Jasiński

---

## [Editor Report · Acceptance letter]

13 Jun 2024

PONE-D-23-00889R1 

PLOS ONE

Dear Dr. Noman, 

I'm pleased to inform you that your manuscript has been deemed suitable for publication in PLOS ONE. Congratulations! Your manuscript is now being handed over to our production team.

Kind regards, 

on behalf of

Professor Jin Liu 

Academic Editor

PLOS ONE